# Pass@K Policy Optimization with Unbiased Advantage Estimation

## Abstract

Reinforcement learning for post-training large language models often relies on sparse, outcome-based rewards, making effective exploration essential. Recently, directly optimizing pass@K (or its generalization, max@K) metric has been shown to promote exploration effectively. In this study, we revisit max@K policy gradient via an alternative formulation and identify a limitation of existing estimators. Although these estimators are unbiased for the policy gradient, they can be biased as advantage estimators, leaving room for further variance reduction through appropriate baseline subtraction. To bridge this gap, we introduce the Leave-Two-Out (L2O) baseline, which provably centers the advantage while preserving unbiasedness of the PG estimator. We further develop a vectorized algorithm for computing L2O, making it practical for large-scale language model training. We validated the practical efficacy of our method on LLM reasoning tasks. Compared to the baseline without L2O, our method achieves a 77.4% average reduction in gradient variance during training on Llama-3.2-3B-Instruct model. Moreover, our method improves average pass@k (k=256) performance by 5.2% on Qwen2.5-Math-7B and 2.4% on Llama-3.2-3B-Instruct across five math reasoning benchmarks.

## 1. Introduction

Reinforcement learning (RL) has emerged as a cornerstone in post-training large language models (LLMs), particularly for complex reasoning tasks (Shao et al., 2024). In these domains, the reward signal is often sparse and outcome-based, awarded only when the final answer is verified as correct. Consequently, effective exploration is paramount to discovering valid solution paths within the vast combinatorial output space.

To promote exploration, prior works have adapted classical RL techniques, such as entropy regularization (Cheng et al., 2025; Zheng et al., 2025b; Cui et al., 2025) and count-based methods (Song et al., 2025a; Bai et al., 2025). These methods typically operate by adding exploration bonuses to the reward. However, balancing these auxiliary objectives with the primary task reward remains a non-trivial challenge.

Recently, as an alternative to adding exploration bonuses, optimizing pass@K, a metric measuring the probability of obtaining at least one correct answer in $K$ independent samples, has been proposed to incentivize answer diversity (Tang et al., 2025; Zheng et al., 2025a; Chen et al., 2025b; Walder & Karkhanis, 2025; Peng et al., 2025; Bagirov et al., 2025; Tuyls et al., 2025). Tang et al. (2025) pioneered the optimization of the pass@K objective and proposed several policy gradient estimators for the case where $K = B$ ($B$ being the sample size of rewards accessible for a given question). Most relevant to our work is Walder & Karkhanis (2025), who proposed a method to compute unbiased policy-gradient estimators for cases where $K \leq B$.

In the literature of policy gradient estimation, reducing the variance of the estimator while maintaining unbiasedness is a critical challenge (Williams, 1992; Greensmith et al., 2004; Mei et al., 2022). A common approach involves using a baseline to ensure that the resulting advantage term has zero expectation (over the action distribution), serving as a proxy for the optimal variance-reduction baseline (Greensmith et al., 2004). Motivated by these considerations, the objective of this work is to propose a centered (zero-mean) advantage term for the pass@K policy gradient (and its known continuous reward generalization, max@K).

However, evaluating the advantage in the naive max@K policy gradient is challenging because the $K$ actions are coupled through the max operator, making it difficult to compute the expectation over the per-action distribution. To address this, we adopt the alternative perspective of the max@K policy gradient proposed by Anonymous (2025), which enables us to measure the *Expected Improvement* (EI) of an action. From this perspective, we observe that the advantage estimators proposed by Walder & Karkhanis (2025) tend to either over-estimate or under-estimate the advantage. Therefore, although their estimators are **unbiased as**

[1]Anonymous Institution, Anonymous City, Anonymous Region, Anonymous Country. Correspondence to: Anonymous Author <anon.email@domain.com>.

Preliminary work. Under review by the International Conference on Machine Learning (ICML). Do not distribute.

**policy-gradient estimators**, they are **biased as advantage estimators**, in that they do not have zero expectation.

To address this issue, we propose the **Leave-Two-Out (L2O)** baseline. The L2O baseline allows us to construct a centered advantage term that reduces variance while preserving unbiasedness of the policy gradient (Sec. 3.1). To ensure practical applicability, a vectorized algorithm is derived to compute the estimator efficiently in $O(B^2)$ time by exploiting the structure of the EI-based formulation (Sec. 3.2). Our method integrates seamlessly with existing RL algorithms like Group Relative Policy Optimization (GRPO) (Shao et al., 2024) as a simple reward transformation (Sec. 3.3).

Empirically, we first validate the variance-reduction effect of our method in controlled bandit and maze environments (Sec. 4). Furthermore, we validate the practical efficacy of our method for LLM reasoning tasks using Llama-3.2-3B and Qwen2.5-Math-7B (Sec. 5). Specifically, our method (EI+L2O) achieved an average reduction of 77.4% in gradient variance during training compared to the existing method without L2O (Walder & Karkhanis, 2025) for Llama-3.2-3B-Instruct model. Our method also improved average pass@$k$ ($k$=256) performance by 5.2% on Qwen2.5-Math-7B and by 2.4% on Llama-3.2-3B-Instruct across five math reasoning benchmarks (AIME24, AIME25, AMC23, MATH500 (Hendrycks et al., 2021), and Minerva (Lewkowycz et al., 2022)). Furthermore, we demonstrated that our method consistently outperforms representative strong baselines (PKPO, GRPO, Entropy-Adv) of LLM post-training methods.

## 2. Background

In this section, we first formalize the setting of reasoning tasks and review the objectives of pass@K and max@K policy optimization. Following this, we outline the motivation for our work and introduce the fundamental concepts required to construct our proposed estimator.

**Setting.** We consider a setting in which an agent generates an action $a \in \mathcal{A}$, where $\mathcal{A}$ is a finite set, and the action is evaluated by a reward function $r : \mathcal{A} \to \mathbb{R}$. Our goal is to learn a policy $\pi_\theta \in \Delta(\mathcal{A})$, parameterized by $\theta \in \mathbb{R}^d$, that maximizes the expected reward $J_{\mathrm{RL}}(\theta) = \mathbb{E}_{a \sim \pi_\theta}[r(a)]$. When it is clear from the context, we omit the dependence on $\pi_\theta$ for brevity. In practice, the policy may take additional information as input (e.g., questions in LLM reasoning), but we omit this dependence, and the same statements naturally apply to that setting (Sec. 3.3). During optimization, we assume access to a batch of $B$ actions $a_1, \ldots, a_B \in \mathcal{A}$ sampled from the policy, along with their corresponding rewards $r_1, \ldots, r_B \in \mathbb{R}$. This collection forms a batch $\mathcal{D} = (a_{1:B}, r_{1:B})$.

### 2.1. Policy Gradient Estimation and Variance Reduction

In this study, we focus on the policy gradient (PG) method, which directly optimizes $\pi_\theta$ via gradient ascent using $\nabla J(\theta) = \mathbb{E}_{a \sim \pi_\theta}[r(a)\nabla_\theta \log \pi_\theta(a)]$ (Williams, 1992) Because policy gradient estimates can exhibit high variance, we introduce a constant baseline $b$ to reduce variance (Greensmith et al., 2004). This yields the following expression.

$$\nabla J(\theta) = \mathbb{E}_{a \sim \pi_\theta}\left[(r(a) - b)\nabla_\theta \log \pi_\theta(a)\right]. \quad (1)$$

We use the term *advantage* to denote the action-dependent scalar multiplier of the score function (i.e., $r(a) - b$ in Eq. (1)). As long as the baseline is action-independent, the PG estimator remains unbiased because $\mathbb{E}_{a \sim \pi_\theta}[b\,\nabla_\theta \log \pi_\theta(a)] = 0$.

A common choice is to use the expected reward as the baseline, $b = \mathbb{E}_{a \sim \pi_\theta}[r(a)]$, so that the advantage has zero mean: $\mathbb{E}_{a \sim \pi_\theta}[r(a) - b] = 0$ (Greensmith et al., 2004). Centering the advantage removes a constant offset from the reward signal, which is often a dominant source of variance.

**Batch estimation.** Given the batch $\mathcal{D}$, we can approximate the gradient using the REINFORCE estimator (Williams, 1992).

$$\hat{g} = \frac{1}{B}\sum_{i=1}^{B}(r_i - b_i)\nabla_\theta \log \pi_\theta(a_i), \quad (2)$$

Here, $b_i$ is a baseline estimate for the $i$-th sample. In this study, we consider two distinct notions of unbiasedness.

1. **Unbiased PG estimator:** We call a PG estimator $\hat{g}$ unbiased when $\mathbb{E}_{\mathcal{D}}[\hat{g}] = \nabla J(\theta)$. Here, $\mathbb{E}_{\mathcal{D}}[\cdot]$ denotes the expectation over the batch $\mathcal{D} = (a_{1:B}, r_{1:B})$ induced by i.i.d. sampling from $\pi_\theta$, and $r$. This requires that the baseline $b_i$ does not depend on $r_i$, which would introduce action dependence into the PG estimator.

2. **Unbiased advantage:** On the other hand, we call the advantage (e.g., $r(a) - b$) unbiased when $\mathbb{E}_{a \sim \pi_\theta}[r(a) - b] = 0$. Likewise, for an advantage estimator (e.g., $r_i - b_i$), unbiasedness requires it to be centered so that $\mathbb{E}_{\mathcal{D}}[r_i - b_i] = 0$. Note that this does not necessarily imply that the batch mean of the baseline is zero, i.e., $\frac{1}{B}\sum_{i=1}^{B} b_i = 0$.

Unbiasedness of the PG estimator is a prerequisite for policy gradient methods, and an unbiased advantage is crucial for variance reduction (Greensmith et al., 2004). Therefore, we aim to achieve both forms of unbiasedness. In Eq. (2), a naive way to obtain an unbiased advantage estimator is to set the baseline to the batch mean reward, $b_i = \frac{1}{B}\sum_{j=1}^{B} r_j$, as an estimator of $\mathbb{E}_{a \sim \pi_\theta}[r(a)]$. However, this violates the independence condition because $b_i$ depends on $r_i$, resulting

in a biased PG estimator. A common remedy is to use a Leave-One-Out (L1O) baseline, $b_i = \frac{1}{B} \sum_{j \neq i} r_j$, to avoid dependence on $r_i$ (Parmas et al., 2018; Mnih et al., 2016). Although L1O works in this simple RL setting, it may fail for more complicated objectives, as is the case for max@K (see Sec. 3.1).

## 2.2. Pass@K and max@K Policy Optimization

Recently, in the context of RL for reasoning tasks with binary rewards (correct/incorrect), directly optimizing the *pass@K* metric has been proposed to incentivize answer diversity (Tang et al., 2025; Zheng et al., 2025a; Chen et al., 2025b; Walder & Karkhanis, 2025; Peng et al., 2025; Bagirov et al., 2025; Tuyls et al., 2025). Pass@K optimization aims to maximize the probability that at least one of $K$ sampled answers is correct. The *max@K* objective generalizes pass@K to continuous reward functions (Walder & Karkhanis, 2025), and is defined as the expected maximum reward among $K \leq B$ samples.

$$J^K(\theta) := \mathbb{E}_{a_{1:K} \sim \pi_\theta} \left[ \max_{k=1,\ldots,K} r(a_k) \right]. \quad (3)$$

The PG for this objective is given by the following expression (Anonymous, 2022).

$$\nabla J^K(\theta) := \mathbb{E}_{a_{1:K}} \left[ \max_{k=1,\ldots,K} r(a_k) \sum_{k=1}^{K} \nabla_\theta \log \pi_\theta(a_k) \right]. \quad (4)$$

Our goal is to propose an estimator that is unbiased as a PG estimator and also yields an advantage with zero expectation. However, the formulation in Eq. (4) makes it difficult to evaluate the advantage because the $K$ actions are coupled through the max operator. To address this issue, we leverage an alternative formulation of the max@K PG.

## 2.3. Expected Improvement Formulation

Anonymous (2025) proposed a method to decouple the max@K PG into a per-action expectation by introducing a baseline $W_{-k} := \max_{k' \neq k} r(a_{k'})$. Using the identity $\max_k r_k - W_{-k} = (r_k - W_{-k})_+$, where $(z)_+ := \max(z, 0)$, the PG can be rewritten as:

$$\nabla J^K(\theta) = \mathbb{E} \left[ \sum_{k=1}^{K} \nabla_\theta \log \pi_\theta(a_k) \max_{k=1,\ldots,K} (r(a_k) - W_{-k}) \right]$$

$$= \mathbb{E} \left[ \sum_{k=1}^{K} \nabla_\theta \log \pi_\theta(a_k) \left( r(a_k) - W_{-k} \right)_+ \right]. \quad$$

The first equality relies on the fact that $W_{-k}$ is independent of $a_k$, making the baseline term vanish in expectation.

*Table 1.* We summarize the properties of the EI estimator versus our L2O baseline. Following the discussion in Sec. 2.1, an unbiased PG estimator is essential for the validity, and an unbiased advantage is critical for minimizing the gradient variance. As discussed in Sec. 2, the EI estimator $s_i$ (Walder & Karkhanis, 2025) is unbiased as a PG estimator, but biased as an advantage estimator. Our L2O baseline ensures unbiasedness both as a PG estimator and as an advantage estimator, achieving further variance reduction.

| Method | PG | Advantage |
|---|---|---|
| EI ($s_i$) | Unbiased | Biased |
| EI + L1O ($s_i - b_{-i}^{\mathrm{L1O}}$) | Biased | Unbiased |
| EI + L2O (ours) ($s_i - b_{-i}^{\mathrm{L2O}}$) | Unbiased | Unbiased |

By exploiting the symmetry under the i.i.d. assumption, we arrive at the simplified form (Prop. 2 of Anonymous (2025)):

$$\nabla J^K(\theta) = K \mathbb{E}_{a \sim \pi_\theta} \left[ \nabla_\theta \log \pi_\theta(a) \, s(a) \right], \quad (5)$$

where the scalar term $s(a)$ is defined as:

$$s(a) := \mathbb{E}_{a_{1:K-1} \sim \pi_\theta} \left[ \left( r(a) - W_{K-1} \right)_+ \right], \quad (6)$$

with $W_{K-1} = \max_{j=1,\ldots,K-1} r(a_j)$. It is referred to as the **Expected Improvement (EI)** of action $a$, as it quantifies the expected gain of $a$ over the best of $K-1$ other samples (Anonymous, 2025).

**Motivation for Further Variance Reduction.** Compared to the standard PG in RL (Eq. (1)), we can see that the EI term $s(a)$ serves as the advantage. Moreover, $s(a)$ corresponds to the expected value of the advantage in the $\max@K - \max@(K-1)$ estimator proposed by Walder & Karkhanis (2025). While their estimator reduces variance compared to naïve approaches, Eq. (6) reveals a critical limitation. Because $s(a)$ is the output of a ReLU, it is nonnegative, implying $\mathbb{E}_{a \sim \pi_\theta}[s(a)] > 0$. Thus, **their estimator is unbiased as a PG estimator, but not in terms of the advantage**, leaving room for further variance reduction by subtracting an appropriate baseline. In Sec. 3, we use this ReLU-based formulation to derive a baseline that ensures the advantage has zero expectation. We also provide an efficient method to compute the resulting estimator.

## 3. Method

We begin in Sec. 3.1 by deriving a novel baseline that theoretically guarantees unbiasedness **both as a PG estimator and as an advantage estimator**. In Sec. 3.2, we propose an efficient method to compute the baseline, reducing the complexity from cubic to quadratic and making it practical for large-scale training. Finally, in Sec. 3.3, we outline how to integrate our approach with modern RL algorithms.

### 3.1. Unbiased Advantage Estimation

Given a batch $\mathcal{D} = (a_{1:B}, r_{1:B})$, we first construct unbiased estimators $s_i$ of the expected improvement $s(a_i)$ defined in Eq. (6).

$$s_i := \mathbb{E}_{\mathcal{I}}\left[\left(r_i - \max_{j \in \mathcal{I}} r_j\right)_+\right]. \qquad (7)$$

Here, $\mathcal{I}$ is a subset of size $K-1$ drawn uniformly *without replacement* from the indices $\{1, \ldots, B\} \setminus \{i\}$. By U-statistics theory (Hoeffding, 1992), $s_i$ is an unbiased estimator of $s(a_i)$. Note that $s_i$ is equivalent to one of Walder & Karkhanis (2025)'s advantage estimators (max@K - max@(K-1)).

Recall the two unbiasedness criteria presented in Sec. 2.1. For the desired estimator, we need to find a baseline $b_i$ that satisfies the following two conditions: (1) **Independence:** $b_i$ does not depend on $r_i$, ensuring an unbiased PG estimator; and (2) **Centering:** the batch expectation of $s_i - b_i$ is zero, ensuring unbiasedness as an advantage estimator.

As discussed in Sec. 2.1, a common approach is the L1O baseline, $b_{-i}^{\text{L1O}} := \frac{1}{B-1} \sum_{j \neq i} s_j$. However, in the max@K setting, the estimator $s_j, j \neq i$ can implicitly depend on $r_i$, because $r_i$ acts as a potential comparator (maximum over the subset of size $K-1$) in the construction of $s_j$. Consequently, $b_{-i}^{\text{L1O}}$ can correlate with $r_i$, violating the independence condition, leading to a biased PG estimator.

**Leave-Two-Out (L2O) Baseline.** To satisfy both conditions, we propose the *Leave-Two-Out* (L2O) baseline:

$$b_{-i}^{\text{L2O}} := \frac{1}{B-1} \sum_{j \neq i} s_j^{(-i)}, \qquad (8)$$

where $s_j^{(-i)}$ denotes the EI estimator for sample $j$ computed over a reduced dataset that excludes sample $i$:

$$s_j^{(-i)} := \mathbb{E}_{\mathcal{I}'}\left[\left(r_j - \max_{k \in \mathcal{I}'} r_k\right)_+\right]. \qquad (9)$$

Here, $\mathcal{I}'$ is a subset of size $K-1$ drawn uniformly from $\{1, \ldots, B\} \setminus \{i, j\}$. We exclude $i$ to ensure independence from $r_i$, while keeping the comparator subset size $K-1$ consistent with the original EI definition and therefore we need to have $K - 1 \leq B - 2$, i.e., $K \leq B - 1$.

By construction, $b_{-i}^{\text{L2O}}$ is independent of $r_i$ because $r_i$ does not appear as a comparator in the computation. The following proposition guarantees the centering condition, showing that the L2O baseline makes the estimator $1/B \sum_{i=1}^{B} \nabla_\theta \log \pi_\theta(a_i)(s_i - b_{-i}^{\text{L2O}})$ which is unbiased in terms of both PG and advantage, as desired.

**Proposition 3.1** (Centering of the L2O Baseline)**.** *For any $i$, the expectation over the batch satisfies:*

$$\mathbb{E}_{\mathcal{D}}\left[s_i - b_{-i}^{\text{L2O}}\right] = 0, \qquad (10)$$

*which implies that $s_i - b_{-i}^{\text{L2O}}$ is centered (zero-mean) as an advantage estimator.*

The proof is provided in Appendix C.1. We summarize the properties of the advantage estimators for max@K PG in Table 1. Furthermore, we provide a unified theoretical view of the max@K estimators including those proposed by Walder & Karkhanis (2025) in Appendix D.

### 3.2. Efficient Batch Computation

While the theoretical properties of the L2O baseline are desirable, its practical application hinges on computational efficiency. Computing $\mathbf{b}^{\text{L2O}} := [b_{-1}^{\text{L2O}}, \ldots, b_{-B}^{\text{L2O}}]^\top$ naively costs $O(B^3)$ because it evaluates $O(B)$ EI terms for each of the $B$ indices $i$, and each EI term naively scans $O(B)$ candidate comparators in the (reduced) batch.

In this section, we propose a vectorized algorithm that reduces this complexity to $O(B^2)$ by leveraging the structure of the **pairwise ReLU differences** in the EI estimator. Notice that the term $\left(r_i - \max_{k \in \mathcal{I}} r_k\right)_+$ can be decomposed based on which sample in the batch acts as the maximum $W = \max_{k \in \mathcal{I}} r_k$. If $r_j$ is the maximum in the subset $\mathcal{I}$ and $r_j < r_i$, the improvement is exactly $r_i - r_j$. If $r_j \geq r_i$, the ReLU zeros out the term. This allows us to replace complex subset iterations with a weighted sum of $(r_i - r_j)_+$ over all $j$, which can be efficiently computed as a matrix multiplication.

**Batch Computation of the EI Estimator.** First, we demonstrate how to efficiently compute the EI vector $\mathbf{s} = [s_{(1)}, \ldots, s_{(B)}]^\top$ using the structure. Assume rewards are sorted in ascending order: $r_{(1)} \leq \cdots \leq r_{(B)}$. For a target sample $i$, the term $\left(r_{(i)} - W\right)_+$ is non-zero only if $W < r_{(i)}$. The probability that the maximum $W$ equals a specific value $r_{(j)}$ (where $j < i$) follows a hypergeometric distribution depending only on the rank $j$:

$$P(W = r_{(j)}) := \frac{\binom{j-1}{K-2}}{\binom{B-1}{K-1}}, \quad \text{for } j < i. \qquad (11)$$

Here, the denominator represents the total ways to choose $K-1$ samples from $B-1$ candidates. The numerator corresponds to fixing $r_{(j)}$ as the maximum and choosing the remaining $K-2$ samples from the $j-1$ candidates smaller than $r_{(j)}$. Consequently, the estimator $s_{(i)}$ can be expressed as a weighted sum of pairwise differences:

$$s_{(i)} = \sum_{j=1}^{B} P(W = r_{(j)})\left(r_{(i)} - r_{(j)}\right)_+, \qquad (12)$$

where we extend the sum to $B$ using the ReLU property $\left(r_{(i)} - r_{(j)}\right)_+ = 0$ for $j \geq i$. This formulation leads directly to a matrix-based computation.

**Theorem 3.2** (Efficient Vectorized Computation of Batch EI). *Let* $\mathbf{r} \in \mathbb{R}^B$ *be the sorted reward vector. Define the ReLU-difference matrix* $D \in \mathbb{R}^{B \times B}$ *as* $D_{i,j} = (r_{(i)} - r_{(j)})_+$, *and the weight vector* $\mathbf{w} \in \mathbb{R}^B$ *as* $w_j = \binom{j-1}{K-2}/\binom{B-1}{K-1}$. *The vector of EI estimators* $\mathbf{s}$ *is given by:*

$$\mathbf{s} = D\mathbf{w}. \tag{13}$$

The proof is provided in Appendix C.2.

**Complexity Analysis.** Our method requires $O(B \log B)$ time for sorting and $O(B^2)$ time to construct the pairwise difference matrix and perform the resulting matrix–vector multiplication. We note that Walder & Karkhanis (2025) derived an algorithm that computes the EI estimator in $O(B \log B + K)$. In the context of LLM training, the batch size $B$ (typically the group size, e.g., 16–64) is small. Moreover, our vectorized formulation maps directly to GPU-friendly dense primitives (e.g., broadcasted elementwise operations and matrix–vector products), making it straightforward to leverage GPU parallelism in practice.

**Batch Computation of the L2O Baseline.** Similar to the EI estimator, we aim to compute the L2O baseline by matrix multiplication. To this end, we rewrite $b_{-i}^{\text{L2O}}$ as a weighted sum of ReLU differences:

$$b_{-i}^{\text{L2O}} = \frac{1}{B-1} \sum_{j \neq i}^{B} \sum_{l \neq i} P(W_{-i} = r_{(l)})(r_{(j)} - r_{(l)})_+ \tag{14}$$

$$= \frac{1}{B-1} \sum_{l \neq i}^{B} P(W_{-i} = r_{(l)}) \sum_{j \neq i} (r_{(j)} - r_{(l)})_+, \tag{15}$$

where $W_{-i}$ is the maximum of a subset drawn from the population excluding $i$. The second equality is due to the fact that the probability of selecting a specific comparator depends only on its rank within the reduced set.

Since $D_{j,l}$ is already computed, we only need to aggregate these values with the correct weights. The inner summation over $j$ resembles a column sum of $D$, excluding the row $i$, which leads to the following efficient computation.

**Theorem 3.3** (Efficient Computation of L2O Baseline). *Let* $S_l = \sum_{k=1}^{B} D_{k,l}$ *be the column sums of* $D$. *Define the Leave-One-Out column mean matrix* $\mathcal{M} \in \mathbb{R}^{B \times B}$ *as* $\mathcal{M}_{i,l} = (S_l - D_{i,l})/(B-1)$. *Define the precomputable L2O weight matrix* $W^{\text{L2O}} \in \mathbb{R}^{B \times B}$ *as:*

$$W_{i,l}^{\text{L2O}} = \begin{cases} 0 & \text{if } l = i, \\ \dfrac{\binom{(l-1)-\mathbb{I}[l>i]}{K-2}}{\binom{B-2}{K-1}} & \text{if } l \neq i. \end{cases} \tag{16}$$

**Algorithm 1** Group-based RL algorithm with EI + L2O
1: **Input:** Policy $\pi_\theta$, the number of questions per batch $M$, group size $G$.
2: **while** not converged **do**
3:     Sample $M$ questions $\{x_m\}_{m=1}^{M}$.
4:     **for** each question $x_m$ **do**
5:         Sample $G$ outputs $\{a_j\}_{j=1}^{G} \sim \pi_\theta(\cdot \mid x_m)$.
6:         Evaluate rewards $\{r_j\}_{j=1}^{G}$ and sort them.
7:         Compute EI scores $\mathbf{s}$ via Theorem 3.2.
8:         Compute the L2O baseline $\mathbf{b}^{\text{L2O}}$ via Theorem 3.3.
9:         Compute advantages: $\mathbf{Adv} = \mathbf{s} - \mathbf{b}^{\text{L2O}}$.
10:     **end for**
11:     Update $\theta$ by maximizing Eq.(18) with $\mathbf{Adv}$.
12: **end while**

*The vector of L2O baselines* $\mathbf{b}^{\text{L2O}} \in \mathbb{R}^B$ *is given by the row-wise dot product:*

$$b_{-i}^{\text{L2O}} = \sum_{l=1}^{B} \mathcal{M}_{i,l} W_{i,l}^{\text{L2O}}. \tag{17}$$

The proof and explicit construction are provided in Appendix C.3. Overall, we can compute the advantage term $s_i - b_{-i}^{\text{L2O}}$ in $O(B^2)$ time, leveraging GPU parallelism. Code for the computation is provided in Appendix B.

### 3.3. Integration with RL Algorithms of LLM training

We integrate our method into modern RL algorithms for LLM post-training. Here, we assume for a question $x \in \mathcal{X}$ ($\mathcal{X}$ is the set of all possible questions), we sample $B$ actions from a policy $\pi_\theta : \mathcal{X} \to \Delta(\mathcal{A})$ and evaluate a reward to form a batch $(a_{1:B}, r_{1:B})$. Given the batch, we can compute EI scores $\mathbf{s}$ and L2O baselines $\mathbf{b}^{\text{L2O}}$ as described in Sec. 3.2. Since the resulting PG estimator $\frac{1}{B} \sum_{i=1}^{B} \nabla_\theta \log \pi_\theta(a_i \mid x)(s_i - b_{-i}^{\text{L2O}})$ is unbiased as PG estimator, our method functions as a reward transformation from $r_{1:B}$ to $s_{1:B} - b_{1:B}^{\text{L2O}}$ that optimizes the max@K objective (Eq. (3)).

While applicable to any PG-based algorithm, we instantiate our method using group-based RL algorithms such as GRPO (Shao et al., 2024) and Dr.GRPO (Liu et al., 2025) for their success in reasoning tasks. Specifically, we partition the batch into $M$ groups such that actions (= output texts) in each group are generated from the same question: $\{(x_1, a_{1:G}^1, r_{1:G}^1), ..., (x_M, a_{1:G}^M, r_{1:G}^M)\}$, where $G = B/M$ is the group size. Each output text consists of a sequence of tokens. Therefore, the advantage for $i$-th token of $j$-th output text in $m$-th group is defined as $A_{j,i}^m = s_{j,i}^m - b_{-j,i}^m$, where $s_{j,i}^m$ and $b_{-j,i}^m$ are group-wise EI scores and L2O baselines, respectively. Note that when calculating this advantage, we usually set the same reward across all tokens in each output text $r_{j,i}^m = r_j^m$, i.e., language models often rely

on sparse, outcome-based rewards in reasoning tasks. Given a reference policy $\pi_{\text{ref}}$, we optimize the following loss:

$$L(\theta) = \frac{1}{M} \sum_{m=1}^{M} \frac{1}{G} \sum_{j=1}^{G} \frac{1}{|a_j^m|} \sum_{i=1}^{|a_j^m|}$$

$$\min\left[ \phi_\theta(x_m, a_{j,i}^m) A_{j,i}^m, \phi_\theta^\varepsilon(x_m, a_{j,i}^m) A_{j,i}^m \right] - \beta \text{KL}(\pi_\theta | \pi_{\text{ref}}), \tag{18}$$

where $\beta \geq 0$, $\varepsilon \geq 0$ are the hyperparameters and KL is the Kullback-Leibler (KL) divergence, $\phi_\theta(x, a_i) = \pi_\theta(a_i \mid x, a_{<i}) / \pi_{\text{ref}}(a_i \mid x, a_{<i})$ is the ratio between our policy and the reference policy, and $\phi_\theta^\varepsilon(x, a_i)$ is the clipped ratio of $\phi_\theta(x, a_i)$ in range of $[1 - \varepsilon, 1 + \varepsilon]$. $|a_j^m|$ is a sequence length of $j$-th text in $m$-th group. Eq. (18) is exactly the same as GRPO (Shao et al., 2024) except for the way of calculating the advantage: While standard GRPO normalizes rewards by dividing by the inner-group standard deviation, a practice that introduces bias into the PG estimator, both our method and the algorithm by Walder & Karkhanis (2025) (Pass@K Policy Optimization; PKPO) avoid this division to preserve the unbiasedness. The key distinction from Walder & Karkhanis (2025) lies in the advantage construction: while their approach is equivalent to using our raw EI scores, we further subtract the L2O baseline to achieve lower variance.

## 4. Toy Experiments

In this section, we empirically validate the efficacy of the proposed L2O baseline.

### 4.1. Bandits

Here, we validate that theoretical properties of the L2O baseline: 1) unbiasedness as PG estimator, 2) variance reduction over the raw EI estimator due to centering the advantage in a bandit setting.

**Setting.** To emulate a discrete-action setting such as LLMs training, we consider a multi-armed bandit setup in which rewards $\mathbf{r} \in \mathbb{R}^{|\mathcal{A}|}$ and logits $\mathbf{l} \in \mathbb{R}^{|\mathcal{A}|}$ are sampled from a standard normal distribution, $\mathcal{N}(0, 1)$. We sample a batch of $B$ actions $a_i \sim \text{softmax}(\mathbf{l})$ and observe their corresponding rewards $r_i$. From each batch, we construct a PG estimator $\hat{g}$ using (1) EI: $s_i$, (2) EI+L2O: $s_i - b_{-i}^{\text{L2O}}$, and (3) EI+L1O: $s_i - b_{-i}^{\text{L1O}}$, as described in Sec. 3. We then sample $N$ independent batches and measure the bias and variance of the estimators $\hat{g}_j$ for $j = 1, \ldots, N$.

To analyze estimator bias, we vary the number of batches $N \in \{10^3, 10^4, 10^5, 10^6\}$ and measure the estimation error $\left\| \frac{1}{N} \sum_{j=1}^{N} \hat{g}_j - g_{\text{true}} \right\|$ to assess whether it decreases as $N$ increases [1]. For the bias plot, we fix $B = 8$ and $K = 2$.

[1] The true gradient $g_{\text{true}}$ is computed using the closed-form

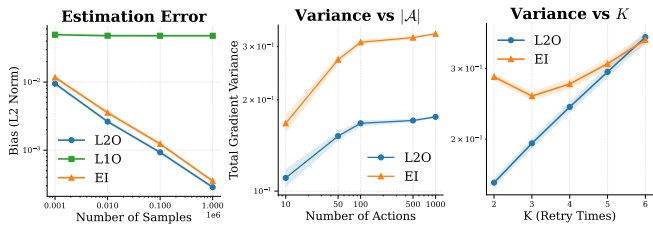

*Figure 1.* The plot of the estimation error (left), variance vs action space size (center), and variance vs $K$ (right). We report the mean and standard error over 100 different random seeds.

We estimate the variance of each estimator via the empirical total variance, $\frac{1}{N} \sum_{j=1}^{N} \|\hat{g}_j - \hat{g}\|^2$, with $N = 10^5$, where $\hat{g} := \frac{1}{N} \sum_{j=1}^{N} \hat{g}_j$ denotes the empirical mean. We investigate the variance-reduction effect from two perspectives: the action space size $|\mathcal{A}|$ and the number of comparators $K$. The gradient variance of the softmax policy can depend on the action space size (App. 7), and in LLMs the action space is typically large. Therefore, we vary $|\mathcal{A}| \in \{10, 50, 100, 1000\}$ to examine how the variance changes, fixing $K = 2$ and $B = 8$. We also study the retry parameter $K$. As $K$ increases, fewer actions tend to have positive expected improvement, which can affect variance. Accordingly, we vary $K \in \{2, 3, 4, 5, 6\}$ while fixing $B = 8$ and $|\mathcal{A}| = 100$. For the detailed explanation of the settings and the additional results, see App. E.1.

**Results.** The results validate the theoretical properties of the L2O baseline. First, Fig. 1 (left) shows that EI+L2O, together with EI, yields a decreasing estimation error that matches the theoretical convergence rate of $\mathcal{O}(1/\sqrt{N})$, indicating that the PG estimator is unbiased. In contrast, EI+L1O exhibits an approximately constant estimation error regardless of the sample size, confirming that the PG estimator is biased.

Regarding variance, Fig. 1 (center) shows that the variance-reduction effect of L2O over raw EI becomes more pronounced as the action space size increases. This trend is particularly relevant for LLMs, where the action space is typically large. Fig. 1 (right) illustrates the impact of $K$. L2O is most effective for moderate $K$ relative to $B = 8$, as it must form $K - 1$ comparators from $B - 2$ samples to maintain unbiasedness. When the comparator set is too small, the baseline estimate becomes noisy, diminishing the variance-reduction effect. These trends hold across various batch sizes (see App. E.1).

### 4.2. Maze Environment

In this section, we utilize a toy maze environment (Anonymous, 2022) to validate both the exploration capability of

derivative of the expected improvement, as described in Proposition 1 of Anonymous (2025).

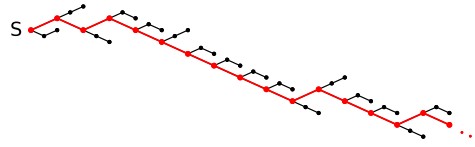

Figure 2. Structure of the Biased Maze Environment. The red line indicates the unique optimal path.

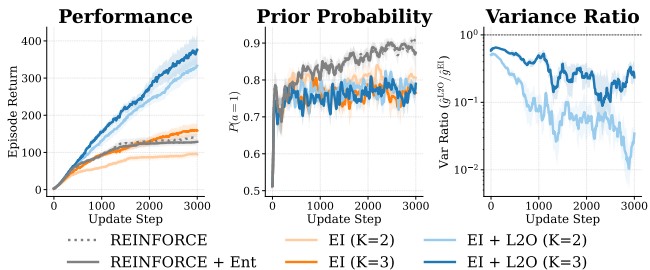

Figure 3. Moving average and the standard error of the average return (left), the probability of action 1 (via global bias $\phi_a$) (center), and the variance (right) over 10 different random seeds. The window size is 10 for the average return and probability and is 200 for the variance ratio.

max@K PG estimators and the variance-reduction effect of the L2O baseline in RL. The maze is deterministic, with binary actions (0 or 1) available at each state. The agent receives a reward of $+1$ for each forward step along a valid path. There exists a single correct path to the goal (indicated by the red line in Fig. 2); if the agent selects the optimal action at every step, it can progress for up to 1000 steps. If the agent takes an incorrect action, the episode terminates within one or two steps. We design the maze to exhibit structural bias, where the correct action is 1 in 75% of the states. Consequently, a policy that blindly prioritizes action 1 can achieve moderately high returns, creating a local optimum that discourages exploration of the optimal path.

**Setting.** We parameterize the policy using tabular logits for each state–action pair, augmented with a global bias vector, $\pi(a \mid s) = \mathrm{softmax}(\theta_{s,a} + \phi_a)$, where $\theta \in \mathbb{R}^{|S| \times |A|}$ and $\phi \in \mathbb{R}^{|A|}$. The global parameter $\phi$ captures an environmental bias toward specific actions. We compare our method (EI+L2O) against three baselines: (1) REINFORCE (Williams, 1992), (2) REINFORCE with entropy regularization, and (3) EI (Walder & Karkhanis, 2025) with $K \in \{2, 3\}$. We collect trajectories with batch size $B = 16$ and train for 3000 iterations. We report the average return, the probability assigned to action 1 by the global bias $\phi$, and the variance ratio of EI+L2O to EI. We use the global bias as a proxy for exploratory tendency, as it represents the default policy in unvisited states (since $\theta$ is initialized near 0). For more details, see App. E.2.

**Results.** The results are shown in Fig. 3. First, we observe that max@K policy gradient estimators (EI and EI+L2O) maintain a more exploratory policy, with the probability of action 1 remaining close to 0.75, compared to the RE-INFORCE baselines, which converge to a greedy strategy with $\Pr(a = 1) > 0.9$. Moreover, EI+L2O substantially outperforms EI, as shown in Fig. 3 (left). We attribute this improvement to the significant variance reduction provided by the L2O baseline (Fig. 3 (right)). Results for other batch sizes and values of $K$ are reported in App. E.2.

## 5. LLM Reasoning Experiments

We evaluate the efficacy of our variance-reduced objective on challenging math reasoning tasks. Our experiments address two questions: (i) does our estimator improve pass@k on standard reasoning benchmarks, and (ii) does it stabilize RL optimization by reducing the variance of the gradient estimator during training.

### 5.1. Experimental Setting

**Training.** We perform RL fine-tuning on Llama-3.2-3B-Instruct (Grattafiori et al., 2024) and Qwen2.5-Math-7B (Yang et al., 2024). We compare our method (EI+L2O) against three baselines: GRPO (Shao et al., 2024), Entropy-Adv (Cheng et al., 2025), and PKPO (Walder & Karkhanis, 2025). Our method follows the algorithm described in Sec. 3.3. We include PKPO as the closest prior method that directly optimizes pass@K/max@K using an unbiased expected-improvement (EI) policy-gradient estimator; in our notation, PKPO can be viewed as the EI-only variant, so this comparison isolates the benefit of our centered advantage estimation. Unless otherwise stated, we use a fixed training objective size $K = 2$. Training data and hyperparameters are provided in App. F.1. Throughout this section, $K$ refers to the training objective size (max@K/pass@K objective), while $k$ denotes the evaluation compute in pass@$k$.

**Evaluation.** We evaluate on five math reasoning benchmarks: AIME24, AIME25, AMC23, MATH500 (Hendrycks et al., 2021), and Minerva (Lewkowycz et al., 2022). All evaluations use nucleus sampling with temperature 0.6 and top-p 0.95. To reduce evaluation variance, we generate $n = 1024$ samples for every benchmark. We report the unbiased pass@$k$ (Chen et al., 2021) metric for $k \in \{1, 2, 4, 8, \ldots\}$, computed as

$$\mathrm{pass@}k := \mathbb{E}_{x \sim \mathcal{D}} \left[ 1 - \frac{\binom{n-c}{k}}{\binom{n}{k}} \right], \qquad (19)$$

where $n$ is the number of sampled completions and $c$ is the number of correct completions among them. For consistent task-averaging in the main text, we report results up to $k \le 256$ for all benchmarks, and additionally report $k \le 1024$ for AIME24, AIME25, and AMC23 in App. F.2.

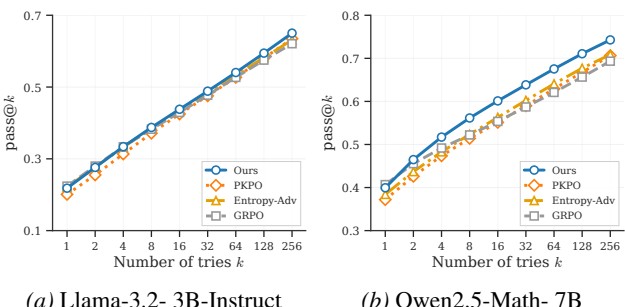

*(a)* Llama-3.2- 3B-Instruct  *(b)* Qwen2.5-Math- 7B

*Figure 4.* Task-average pass@$k$ ($k \leq 256$). Unweighted average over AIME24, AIME25, AMC23, MATH500, and Minerva (temperature 0.6, top-p 0.95). Our method demonstrates consistent improvement over strong baselines.

### 5.2. Main Results

Figure 4 presents the task-average pass@$k$ up to $k = 256$ for both Llama and Qwen. Across benchmarks, our method (EI+L2O) consistently outperforms all baselines, including PKPO and GRPO. Specifically, compared to PKPO, our method improved average pass@$k$ ($k$=256) performance by 5.2% on Qwen2.5-Math-7B and by 2.4% on Llama-3.2-3B-Instruct across five math reasoning benchmarks. The improvements persist across $k$, indicating that variance reduction helps the policy better exploit additional samples at inference time. We provide extended results up to $k = 1024$ for AIME24, AIME25, and AMC23 in App. F.2.

### 5.3. Effect of Objective Size $K$

We study the impact of the training objective size $K$ by comparing $K \in \{2, 4\}$ while keeping other settings fixed. As shown in Figure 5, increasing $K$ consistently decreases pass@1 but improves pass@$k$ for large $k$ (e.g., $k = 256$) for both Llama and Qwen. This indicates an exploratio-exploitation trade-off: increasing $K$ promotes wider exploration, which reduces one-shot accuracy but improves the probability of success when multiple samples are available. Overall, $K = 2$ offers the best balance in our setting and is used as the default.

### 5.4. Variance Reduction During Training

To test our theoretical motivation, we measure the variance of model gradients during RL training on Llama-3.2-3B-Instruct. Directly computing gradient variance from raw gradients is expensive at LLM scale, so we estimate it from the Adam optimizer (Kingma & Ba, 2017) states. Specifically, Adam maintains exponential moving averages of the first and second moments of gradients; after bias correction, we estimate the element-wise variance as $\mathrm{Var}(g) \approx \hat{v}_t - \hat{m}_t^2$, and aggregate it across all parameters by summing.

Figure 6 reports mean $\pm$ standard deviation over three ran-

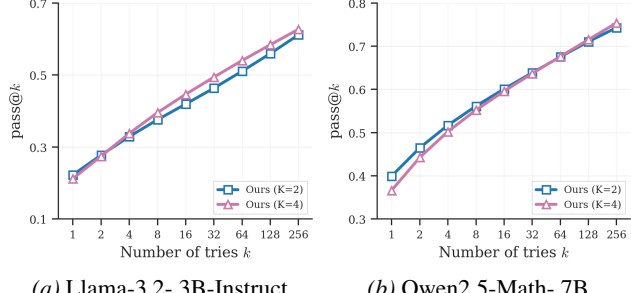

*(a)* Llama-3.2- 3B-Instruct  *(b)* Qwen2.5-Math- 7B

*Figure 5.* Ablation on the training objective size $K$ ($K \in \{2, 4\}$). For both Llama and Qwen, increasing $K$ lowers pass@1 while boosting pass@$k$ for large $k$, indicating an exploration-exploitation trade-off: broader exploration reduces one-shot accuracy but improves success probability given more samples.

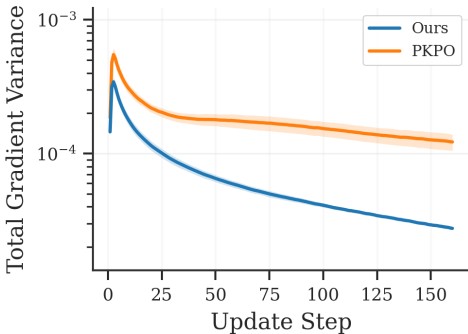

*Figure 6.* Gradient variance during training (3 seeds). Gradient variance is estimated from Adam states via $\mathrm{Var}(g) \approx \hat{v}_t - \hat{m}_t^2$ and aggregated across parameters. Our method (EI+L2O) reduces gradient variance compared to PKPO across training, supporting the proposed variance-reduction mechanism.

dom seeds. Our method yields consistently lower gradient variance than PKPO throughout training, indicating more stable policy gradient updates. Notably, at the end of the training, our method achieved an average reduction of 77.4% in gradient variance compared to PKPO. This provides direct empirical evidence that the L2O baseline reduces estimator variance in practice.

## 6. Conclusion

In this study, we re-examined policy gradient optimization for pass@$K$ and max@$K$ objectives through the lens of expected improvement. Our analysis revealed that existing advantage estimators suffer from non-zero mean values, which induces high variance in the policy gradient estimates. To mitigate this, we proposed the Leave-Two-Out (L2O) baseline, which ensures both an unbiased policy gradient and a zero-mean advantage. We also developed an efficient, vectorized implementation of L2O suitable for large-scale training. Empirical results on LLM reasoning tasks demonstrated that our approach significantly reduces variance and improves overall performance.

## Impact Statement

This paper presents work whose goal is to advance the field of Machine Learning. There are many potential societal consequences of our work, none which we feel must be specifically highlighted here.

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

# A. Additional Related Work

## A.1. RLVR in LLMs

Reinforcement learning with verifiable rewards (RLVR) (Lambert et al., 2025; Guo et al., 2025) is a training paradigm that optimizes LLM policies in domains such as math and code using deterministic feedback from objective verifiers to ground model reasoning in provable correctness rather than subjective human preference (Christiano et al., 2017). Despite this promise (Guo et al., 2025; Jaech et al., 2024), recent studies suggest that RLVR in LLMs primarily serves to amplify behaviors that are already present in the base model (Liu et al., 2025; Zhao et al., 2025; Shah et al., 2025). Yue et al. (2025) investigated the pass@K metric (Chen et al., 2021; Song et al., 2025b; Dang et al., 2025; Wen et al., 2025; Wu et al., 2025), which measures the probability that at least one correct solution is obtained when drawing K independent samples (i.e., Best of K), and experimentally found that as k increases, the base model's pass@K eventually exceeds that of the model trained with RLVR. This phenomenon has been linked to diversity collapse (Dang et al., 2025; Cui et al., 2025) and squeezing reasoning paths (Matsutani et al., 2025; Hu et al., 2025; Bu et al., 2025). Several studies (Wang et al., 2025; Zhang et al., 2025b; Chen et al., 2025a; Zhang et al., 2025a) insist the importance of mid-training as is evident from empirical results, RLVR underperforms with Llama (Grattafiori et al., 2024) compared to Qwen (Yang et al., 2025b;a).

## A.2. Exploration in RL for LLM

In light of these limitations, prior studies have incorporated exploration into RLVR. Cui et al. (2025); Cheng et al. (2025); Zheng et al. (2025b); Shen (2025); Jiang et al. (2025) leveraged entropy bonus to encourage exploration via policy uncertainty. Song et al. (2025a) proposed outcome-based exploration by utilizing UCB-style bonus (Auer et al., 2002). Yu et al. (2025) applied self-penalization by assigning negative rewards to high-confidence answers that deviate from the majority consensus. He et al. (2025) incorporated reward by up-weighting low-probability but correct trajectories, and Gao et al. (2025) adopted Random Network Distillation (RND) (Burda et al., 2018) to provide bonuses for unknown trajectories. Zhou et al. (2025) augmented a semantic-based novelty score calculated with embeddings, Li et al. (2025) employed a semantic-based diversity score with external semantic comparator, and Tuyls et al. (2025) computed a representation-based novelty score with hidden states to boost exploration. Liang et al. (2025) leveraged gradients of reward models to improve temperature sampling. Setlur et al. (2025) promoted in-context exploration via skill asymmetries and negative gradients, enabling reliable extrapolation with increased test-time compute.

Other studies directly optimize Pass@K metric; these are discussed in App. A.4

## A.3. Policy Gradient Estimator and Baseline

The policy gradient is the central approach to optimize the policy in RL (Williams, 1992; Greensmith et al., 2004; Parmas, 2018; Parmas & Sugiyama, 2021). The high variance of gradient estimates is a fundamental challenge in policy gradient methods, often hindering stable convergence (Williams, 1992). To mitigate this, the method of control variates—subtracting a baseline $b(x)$ from the return—is the standard variance reduction technique. Theoretically, the optimal baseline depends on the norm of the score function and the specific reward structure (Peters & Schaal, 2008; Weaver & Tao, 2013). Greensmith et al. (2004) provided a comprehensive analysis, demonstrating that while the exact optimal baseline is computationally expensive, a baseline that approximates the expected return—thereby *centering* the advantage function—captures the majority of the variance reduction benefits. This insight justifies the widespread adoption of value function baselines in modern algorithms like PPO (Schulman et al., 2017) and A3C (Mnih et al., 2016).

In domains involving discrete latent variables or sequence generation, where learning a separate value function is often unstable or costly, *sample-based baselines* have become the dominant approach. This concept was further refined in the context of variational inference by Mnih & Rezende (2016) (VIMCO) and Gu et al. (2015), which utilize the average reward of other samples in the batch (Leave-One-Out; L1O) to construct a low-variance gradient estimator for discrete variables. Specifically, Tucker et al. (2017) highlighted that such control variates are essential for training effective estimators in high-dimensional discrete spaces.

Recently, these multi-sample baseline techniques have been adapted for reasoning tasks in LLMs. Group Relative Policy Optimization (GRPO) (Shao et al., 2024) applies group-based normalization, effectively an L1O baseline, to stabilize training without a critic network. However, these standard L1O techniques rely on the linearity of the expectation operator. Wu et al. (2018) discussed the complexities of action-dependent baselines, but the specific challenges of the *max@K* objective remain underexplored. In the max@K setting, the non-linear dependency between samples introduced by the max

operator renders standard L1O baselines biased, necessitating the development of our strictly unbiased Leave-Two-Out (L2O) approach.

### A.4. Pass@ K Policy Optimization

Recent advancements in reasoning tasks have increasingly focused on directly optimizing the *pass@K* metric, an evaluation criterion where $K$ independent samples are drawn from the model, and success is defined by at least one sample being correct (Tang et al., 2025; Walder & Karkhanis, 2025; Peng et al., 2025; Bagirov et al., 2025; Chen et al., 2025b). The primary objective of this approach is to maintain model diversity, thereby encouraging exploration to discover correct solutions. Tang et al. (2025) first proposed optimizing the pass@K objective using RL, while Chen et al. (2025b) provided an empirical analysis of its effects on model behavior, particularly regarding entropy and generation diversity. Most relevant to our work is Walder & Karkhanis (2025), which generalizes the pass@K objective to continuous rewards (denoted as *max@K*) and proposes multiple estimators for this objective. However, we observe that the estimators proposed by Walder & Karkhanis (2025) can lead to over-estimated advantage terms. To address this, we re-examine the max@K objective and propose a novel estimator designed to ensure the advantage function has an expected value of zero, thereby stabilizing optimization.

Prior to the development of pass@K policy optimization, the RL community has explored optimizing expected rewards across multiple trials (Anonymous, 2022). Anonymous (2022) formally introduced the *ReMax* objective to maximize the max@K outcome, originally optimizing it within a resettable simulator. Subsequently, Anonymous (2025) extended this framework by deriving a policy gradient estimator. We are inspired by the concept they proposed, the expected improvement form of the policy gradient estimator, which enables us to analyze the advantage term of the max@K policy gradient. However, a direct application is infeasible in the language model setting, where computing rewards for the entire action space is intractable. Therefore, we propose a practical estimator that approximates the EI using only a limited batch of samples.

## B. Code to Compute the Estimators

Below is the code to compute the statistics proposed in Sec. 3. That is, EI $s_i$ and L2O baseline $b_{-i}^{L2O}$.

```python
import jax.numpy as jnp
from scipy.special import gammaln

def comb(n, k):
    """Computes binomial coefficient C(n, k) in log-space."""
    return jnp.exp(gammaln(n + 1) - gammaln(k + 1) - gammaln(n - k + 1))

def compute_batch_ei(returns: jnp.ndarray, K: int) -> jnp.ndarray:
    """
    Computes the unbiased EI estimator s_i using matrix operations.
    Corresponds to Theorem 3.2.
    """
    B = returns.shape[0]
    # Sort rewards: r_{(1)} <= ... <= r_{(B)}
    order = jnp.argsort(returns)
    r_sorted = returns[order]

    # Compute Difference Matrix D_{i,j} = (r_{(i)} - r_{(j)})_+
    D = jnp.maximum(r_sorted[:, None] - r_sorted[None, :], 0.0)

    # Compute Weight Vector w_j (Theorem 3.2)
    # w_j = C(j-1, K-2) / C(B-1, K-1)
    ranks = jnp.arange(B)
    w = comb(ranks, K - 2) / comb(B - 1, K - 1)
    w = jnp.nan_to_num(w) # Handle cases where rank < K-2
    # s = D @ w
    s_sorted = D @ w
    # Restore original order
    return s_sorted[jnp.argsort(order)]

def compute_l2o_baseline(returns: jnp.ndarray, K: int) -> jnp.ndarray:
    """
    Computes the L2O baseline b_{-i}^{L2O} (Theorem 3.3) in O(B^2).
    """
    B = returns.shape[0]
    order = jnp.argsort(returns)
    r_sorted = returns[order]

    # Difference Matrix D and Column Sums S_l
    D = jnp.maximum(r_sorted[:, None] - r_sorted[None, :], 0.0)
    S = jnp.sum(D, axis=0) # S_l = sum_k D_{k,l}

    # LOO Column Mean Matrix M_{i,l} (Theorem 3.3)
    # M_{i,l} = (S_l - D_{i,l}) / (B - 1)
    M = (S[None, :] - D) / (B - 1)

    # L2O Weight Matrix W^{L2O}_{i,l}
    # Rank adjustment: (l-1) - 1 if l > i else (l-1)
    i_idx = jnp.arange(B)[:, None]
    l_idx = jnp.arange(B)[None, :]
    adjusted_rank = l_idx - (l_idx > i_idx).astype(jnp.float32)

    W = comb(adjusted_rank, K - 2) / comb(B - 2, K - 1)
    W = jnp.nan_to_num(W)
    W = W * (1.0 - jnp.eye(B)) # Set diagonal (l=i) to 0

    # b_{-i}^{L2O} = sum_l M_{i,l} * W_{i,l}
    b_sorted = jnp.sum(M * W, axis=1)
    return b_sorted[jnp.argsort(order)]
```

*Listing 1.* Code to compute the statistics proposed in Sec. 3

## C. Proofs

### C.1. Proof of Proposition 3.1

In this appendix, we provide a formal proof for Proposition 3.1. We show that both the EI estimator $s_i$ and the L2O baseline $b_{-i}^{L2O}$ are unbiased estimators of the same population quantity, ensuring the centered advantage property.

***Proof.*** We assume $2 \leq K \leq B-1$ so that all subsets used below are well-defined. Let $\mathcal{D} = (a_{1:B}, r_{1:B})$ be a set of samples where each action is drawn i.i.d. from $\pi_\theta$ and $r_i := r(a_i)$. We define the *population Expected Improvement* $\mu_{EI}$ as the expected gain of a single action against the maximum of $K-1$ other independent samples. Formally, let $a_1, \ldots, a_K \overset{\text{i.i.d.}}{\sim} \pi_\theta$ be $K$ independent random variables. Due to the i.i.d. assumption on $a_1, \ldots, a_K$, the population EI is defined as:

$$\mu_{EI} := \mathbb{E}_a\left[s(a)\right] = \mathbb{E}_{a_{1:K}}\left[\left(r(a_1) - \max_{k=2,\ldots,K} r(a_k)\right)_+\right]. \tag{20}$$

**Unbiasedness of the EI Estimator** $s_i$. First, we verify that $s_i$ targets $\mu_{EI}$ following the U-statistics theory (Hoeffding, 1992). Given a batch $\mathcal{D} = (a_{1:B}, r_{1:B})$, $s_i$ is constructed by averaging over all possible subsets of size $K-1$ from the batch excluding $i$ (denoted as $\mathcal{U}_{-i} = \{1, \ldots, B\} \setminus \{i\}$).

$$s_i := \mathbb{E}_{\mathcal{I}}\left[\left(r_i - \max_{k\in\mathcal{I}} r_k\right)_+\right] = \frac{1}{\binom{B-1}{K-1}} \sum_{\mathcal{I}\subseteq\mathcal{U}_{-i}, |\mathcal{I}|=K-1} \left(r_i - \max_{k\in\mathcal{I}} r_k\right)_+. \tag{21}$$

Taking the expectation over the batch $\mathcal{D}$:

$$\mathbb{E}_{\mathcal{D}}\left[s_i\right] = \frac{1}{\binom{B-1}{K-1}} \sum_{\mathcal{I}\subseteq\mathcal{U}_{-i}, |\mathcal{I}|=K-1} \mathbb{E}_{\mathcal{D}}\left[\left(r_i - \max_{k\in\mathcal{I}} r_k\right)_+\right]. \tag{22}$$

For any fixed subset $\mathcal{I}$ of size $K-1$, the set of indices $\{i\}\cup\mathcal{I}$ constitutes $K$ distinct samples. Again, by the i.i.d. assumption, the joint distribution of $\{a_i\} \cup \{a_k\}_{k\in\mathcal{S}}$ depends only on the number of samples, not their indices. Thus:

$$\mathbb{E}_{\mathcal{D}}\left[\left(r_i - \max_{k\in\mathcal{I}} r_k\right)_+\right] = \mathbb{E}_{a_{1:K}}\left[\left(r(a_1) - \max_{k=2,\ldots,K} r(a_k)\right)_+\right] = \mu_{EI}. \tag{23}$$

Substituting this back:

$$\mathbb{E}_{\mathcal{D}}\left[s_i\right] = \frac{1}{\binom{B-1}{K-1}} \sum_{\mathcal{I}\subseteq\mathcal{U}_{-i}, |\mathcal{I}|=K-1} \mu_{EI} = \mu_{EI}. \tag{24}$$

**Unbiasedness of the L2O Baseline** $b_{-i}^{L2O}$. Next, we show that the baseline also targets $\mu_{EI}$. The L2O baseline is defined as the average of leave-two-out estimators:

$$b_{-i}^{L2O} := \frac{1}{B-1} \sum_{j\neq i} s_j^{(-i)}. \tag{25}$$

Here, $s_j^{(-i)}$ is the EI estimator for sample $j$ computed using indices $\mathcal{U}_{-ij} = \{1, \ldots, B\} \setminus \{i, j\}$. Crucially, to estimate the same quantity $\mu_{EI}$ (which involves a 1-vs-$(K-1)$ comparison), $s_j^{(-i)}$ must aggregate over subsets of size $K-1$:

$$s_j^{(-i)} := \frac{1}{\binom{B-2}{K-1}} \sum_{\mathcal{I}'\subseteq\mathcal{U}_{-ij}, |\mathcal{I}'|=K-1} \left(r_j - \max_{k\in\mathcal{I}'} r_k\right)_+. \tag{26}$$

By the linearity of expectation:

$$\mathbb{E}_{\mathcal{D}}\left[b_{-i}^{L2O}\right] = \frac{1}{B-1} \sum_{j\neq i} \mathbb{E}_{\mathcal{D}}\left[s_j^{(-i)}\right]. \tag{27}$$

Focusing on a single term $\mathbb{E}_{\mathcal{D}}\left[s_j^{(-i)}\right]$:

$$\mathbb{E}_{\mathcal{D}}\left[s_j^{(-i)}\right] = \frac{1}{\binom{B-2}{K-1}} \sum_{\mathcal{I}' \subseteq \mathcal{U}_{-ij}, |\mathcal{I}'|=K-1} \mathbb{E}_{\mathcal{D}}\left[\left(r_j - \max_{k \in \mathcal{I}'} r_k\right)_+\right]. \tag{28}$$

Similar to the proof of the unbiasedness of the EI estimator, for any subset $\mathcal{I}'$ of size $K-1$ drawn from $\mathcal{U}_{-ij}$, the set $\{j\} \cup \mathcal{I}'$ consists of $K$ distinct i.i.d. samples. Thus, by the i.i.d. assumption, we have:

$$\mathbb{E}_{\mathcal{D}}\left[\left(r_j - \max_{k \in \mathcal{I}'} r_k\right)_+\right] = \mathbb{E}_{a_{1:K}}\left[\left(r(a_1) - \max_{k=2,\ldots,K} r(a_k)\right)_+\right] = \mu_{EI}. \tag{29}$$

Summing over all subsets:

$$\mathbb{E}_{\mathcal{D}}\left[s_j^{(-i)}\right] = \frac{1}{\binom{B-2}{K-1}} \cdot \binom{B-2}{K-1} \cdot \mu_{EI} = \mu_{EI}. \tag{30}$$

Finally, averaging over $j \neq i$:

$$\mathbb{E}_{\mathcal{D}}\left[b_{-i}^{L2O}\right] = \frac{1}{B-1} \sum_{j \neq i} \mu_{EI} = \mu_{EI}. \tag{31}$$

### C.1.1. CONCLUSION

Combining Eq. (24) and Eq. (31), we obtain:

$$\mathbb{E}_{\mathcal{D}}\left[s_i - b_{-i}^{L2O}\right] = \mu_{EI} - \mu_{EI} = 0. \tag{32}$$

This completes the proof. □

### C.2. Proof of Theorem 3.2

Here we prove Theorem 3.2, which shows that the EI estimator can be computed efficiently using matrix operations.

*Proof.* Assume $2 \leq K \leq B$. Let the batch rewards be sorted as $r_{(1)} \leq \cdots \leq r_{(B)}$, and fix a rank index $i \in \{1, \ldots, B\}$. By definition,

$$s_{(i)} := \mathbb{E}_{\mathcal{I}}\left[\left(r_{(i)} - \underbrace{\max_{j \in \mathcal{I}} r_{(j)}}_{:=W}\right)_+\right], \tag{33}$$

$$= \sum_{j=1}^{B} P(W = r_{(j)})\left(r_{(i)} - r_{(j)}\right)_+, \tag{34}$$

where the probability is for the distribution of the uniformly sampled subset $\mathcal{I}$ of size $K-1$ from the comparator pool $\mathcal{U}_{-i} = \{1, \ldots, B\} \setminus \{i\}$. Since $(r_{(i)} - r_{(j)})_+ = 0$ for all $j \geq i$, only ranks $j < i$ can contribute to the expectation. Therefore, it suffices to compute $P(W = r_{(j)})$ for $j < i$.

Fix $j < i$. The event $\{W = r_{(j)}\}$ occurs if and only if $j \in \mathcal{I}$ and the remaining $K-2$ indices in $\mathcal{I}$ are chosen from the $j-1$ indices with rewards strictly smaller than $r_{(j)}$. The number of such subsets is $\binom{j-1}{K-2}$. The total number of comparator subsets is $\binom{B-1}{K-1}$. Hence, for $j < i$,

$$P(W = r_{(j)}) = \frac{\binom{j-1}{K-2}}{\binom{B-1}{K-1}}. \tag{35}$$

Using the law of total expectation and the fact that only $j < i$ contributes, we obtain

$$s_{(i)} = \sum_{j=1}^{B} P(W = r_{(j)})\left(r_{(i)} - r_{(j)}\right)_+ = \sum_{j=1}^{B} w_j\, D_{i,j}, \tag{36}$$

where $D_{i,j} := (r_{(i)} - r_{(j)})_+$ and $w_j := \binom{j-1}{K-2}/\binom{B-1}{K-1}$. Stacking the identities for all $i$ yields $\mathbf{s} = D\mathbf{w}$. □

### C.3. Proof of Theorem 3.3

Here we prove Theorem 3.3, which shows that the L2O baseline can be computed efficiently using matrix operations.

*Proof.* Assume $2 \leq K \leq B - 1$. Let $r_{(1)} \leq \cdots \leq r_{(B)}$ be the sorted rewards. Define $D \in \mathbb{R}^{B \times B}$ by $D_{j,l} := \left(r_{(j)} - r_{(l)}\right)_+$. Fix an index $i \in \{1, \ldots, B\}$. Recall that

$$b_{-i}^{\text{L2O}} := \frac{1}{B-1} \sum_{j \neq i} s_j^{(-i)}. \tag{37}$$

For each $j \neq i$, define $s_j^{(-i)}$ by sampling $\mathcal{I}'$ uniformly without replacement among all subsets of size $K - 1$ from $\{1, \ldots, B\} \setminus \{i, j\}$ and setting $W_{-i} := \max_{k \in \mathcal{I}'} r_{(k)}$. Then

$$s_j^{(-i)} = \mathbb{E}_{\mathcal{I}'}\left[\left(r_{(j)} - W_{-i}\right)_+\right] = \sum_{l \in \{1, \ldots, B\} \setminus \{i, j\}} P\left(W_{-i} = r_{(l)}\right) \left(r_{(j)} - r_{(l)}\right)_+. \tag{38}$$

Note that $D_{j,l} = (r_{(j)} - r_{(l)})_+ = 0$ for all $l \geq j$. For $l \geq j$, we have $D_{j,l} = (r_{(j)} - r_{(l)})_+ = 0$, hence these terms do not contribute regardless of the value assigned to $W_{i,l}^{\text{L2O}}$. Therefore, it suffices to compute $P(W_{-i} = r_{(l)})$ for $l < j$.

Fix $l < j$ with $l \neq i$. The event $\{W_{-i} = r_{(l)}\}$ occurs if and only if $l \in \mathcal{I}'$ and the remaining $K - 2$ indices in $\mathcal{I}'$ are chosen from indices with rank strictly below $l$. Among $\{1, \ldots, l-1\}$, the only index that may be excluded from the sampling pool $\{1, \ldots, B\} \setminus \{i, j\}$ is $i$ when $i < l$. Since $l < j$, excluding $j$ does not remove any index from $\{1, \ldots, l-1\}$. Thus, the number of available indices strictly below $l$ in the pool is $(l-1) - \mathbb{I}[l > i]$. Hence, the number of valid subsets $\mathcal{I}'$ of size $K - 1$ for which $W_{-i} = r_{(l)}$ is $\binom{(l-1) - \mathbb{I}[l > i]}{K - 2}$. The total number of possible subsets $\mathcal{I}'$ is $\binom{B-2}{K-1}$. Therefore, for any $l < j$ with $l \neq i$,

$$P\left(W_{-i} = r_{(l)}\right) = \frac{\binom{(l-1) - \mathbb{I}[l > i]}{K - 2}}{\binom{B-2}{K-1}} = W_{i,l}^{\text{L2O}}. \tag{39}$$

Crucially, the right-hand side depends on $i$ and $l$ but not on $j$, as long as $l < j$.

Substituting (39) into (38) and using $D_{j,l} = (r_{(j)} - r_{(l)})_+$ yields

$$s_j^{(-i)} = \sum_{l=1}^{B} W_{i,l}^{\text{L2O}} D_{j,l}. \tag{40}$$

Averaging over $j \neq i$ and exchanging the finite sums gives

$$b_{-i}^{\text{L2O}} = \frac{1}{B-1} \sum_{j \neq i} \sum_{l=1}^{B} W_{i,l}^{\text{L2O}} D_{j,l} = \sum_{l=1}^{B} W_{i,l}^{\text{L2O}} \frac{1}{B-1} \sum_{j \neq i} D_{j,l}. \tag{41}$$

Let $S_l := \sum_{k=1}^{B} D_{k,l}$. Then

$$\frac{1}{B-1} \sum_{j \neq i} D_{j,l} = \frac{S_l - D_{i,l}}{B-1} =: \mathcal{M}_{i,l}. \tag{42}$$

Substituting into (41) yields

$$b_{-i}^{\text{L2O}} = \sum_{l=1}^{B} \mathcal{M}_{i,l} W_{i,l}^{\text{L2O}}. \tag{43}$$

This holds for each $i \in \{1, \ldots, B\}$, which completes the proof. $\square$

## D. Unified View of Estimators

In this appendix, we provide a unified view of our proposed estimators and the estimators proposed by Walder & Karkhanis (2025) in terms of the marginal statistics perspective.

## D.1. Definitions of Marginal Statistics

Let $\mathcal{B} = \{1, \ldots, B\}$ be the set of indices for the batch of samples. We assume a fixed batch size $B$ and a subset size $K$ such that $2 \leq K < B - 1$. For any subset $S \subseteq \mathcal{B}$, let $M(S) := \max_{j \in S} r_j$. All expectations below are finite averages over subsets, equivalent to sampling uniformly without replacement.

We formally define the two key statistics $u_i$ and $v_i$ for a specific index $i \in \mathcal{B}$:

- **Conditional Expected Max** $u_i$: The expected maximum of a size-$K$ group conditioned on containing $i$:

$$u_i := \frac{1}{\binom{B-1}{K-1}} \sum_{S \subseteq \mathcal{B} \setminus \{i\}, |S| = K-1} \max(r_i, M(S)). \tag{44}$$

- **Leave-One-Out Expected Max** $v_i$: The expected maximum of a size-$K$ group drawn from the pool excluding $i$:

$$v_i := \frac{1}{\binom{B-1}{K}} \sum_{T \subseteq \mathcal{B} \setminus \{i\}, |T| = K} M(T). \tag{45}$$

The quantity $\gamma(u_i - v_i)$, where $\gamma = K/B$, represents the exact marginal contribution of sample $i$ to the max@K objective under finite without-replacement sampling.

## D.2. Relationship Between EI-L2O and Marginal Statistics

We now show that our bias-corrected signal $\gamma(s_i - b_i)$ is identical to the marginal contribution $\gamma(u_i - v_i)$. First, we explicitly define the finite-batch Expected Improvement $s_i$ and the L2O baseline $b_i$:

- **Expected Improvement** $s_i$: The direct finite-sample version of $s_i = \mathbb{E}[(r_i - W)_+]$ where $W$ is the maximum of $K - 1$ comparators:

$$s_i := \frac{1}{\binom{B-1}{K-1}} \sum_{S \subseteq \mathcal{B} \setminus \{i\}, |S| = K-1} (r_i - M(S))_+. \tag{46}$$

- **L2O Baseline** $b_i$: The average of leave-two-out EI values:

$$b_i := \frac{1}{B-1} \sum_{j \neq i} s_j^{(-i)}, \tag{47}$$

  where $s_j^{(-i)}$ is the EI of $j$ computed over the pool $\mathcal{B} \setminus \{i, j\}$.

Using the identity $\max(r_i, M(S)) = M(S) + (r_i - M(S))_+$, we decompose $u_i$ as:

$$u_i = w_i + s_i, \tag{48}$$

where $w_i$ is the expected comparator-max when sampling $K - 1$ items from the pool excluding $i$. Crucially, the L2O baseline $b_i$ removes this comparator-max contribution. It can be shown through combinatorial identity that:

$$b_i = v_i - w_i. \tag{49}$$

Combining these decompositions yields:

$$s_i - b_i = (u_i - w_i) - (v_i - w_i) = u_i - v_i. \tag{50}$$

Thus, the scaled signal $\gamma(s_i - b_i)$ exactly recovers the marginal contribution $\gamma(u_i - v_i)$.

## D.3. Comparison with Walder & Karkhanis (2025)'s estimators

Walder & Karkhanis (2025) provide two primary advantage estimators: `sloo_minus_one` (their Eq. 33) and `sloo` (their Eq. 29). Here, we formaly show the equivalence of the EI and their `sloo_minus_one` estimator (max@K - max@(K-1)). Then, we describe how the other `sloo` estimator is biased in different way than `sloo_minus_one`.

The sloo_minus_one variant is defined as the difference between the max of a size-$K$ subset and the max of the same subset excluding index $i$.

$$s_i^{loo-1} = \frac{1}{\binom{B}{K}} \sum_{S \subseteq \mathcal{B} \setminus \{i\}, |S|=K-1} \left( \max(r_i, M(S)) - M(S) \right). \tag{51}$$

Using the fundamental identity relating the maximum to the ReLU (Positive Part) function, $\max(a, b) = b + (a - b)_+$, the term inside the summation becomes:

$$\max(r_i, M(S)) - M(S) = (r_i - M(S))_+. \tag{52}$$

Substituting this back into the estimator and multiplying by $\binom{B-1}{K-1} / \binom{B-1}{K-1}$ to align with the definition of Expected Improvement ($s_i$):

$$s_i^{loo-1} = \frac{\binom{B-1}{K-1}}{\binom{B}{K}} \left[ \frac{1}{\binom{B-1}{K-1}} \sum_{S \subseteq \mathcal{B} \setminus \{i\}, |S|=K-1} (r_i - M(S))_+ \right] \tag{53}$$

$$= \frac{K}{B} s_i = \gamma s_i. \tag{54}$$

This proves that Walder's sloo_minus_one is exactly equivalent to our EI-only signal scaled by $\gamma$.

Walder & Karkhanis (2025)'s sloo variant is defined as:

$$s_i^{loo} = S(i, K, \mathcal{B}) - \frac{1}{B-1} \sum_{j \in \mathcal{B} \setminus \{i\}} S(j, K, \mathcal{B} \setminus \{i\}). \tag{55}$$

Walder & Karkhanis (2025) define $S(i, K, \mathcal{U})$ as a normalized sum over $K$-subsets containing index $i$:

$$S(i, K, \mathcal{U}) := \frac{1}{\binom{|\mathcal{U}|}{K}} \sum_{I \subseteq \mathcal{U}, |I|=K, i \in I} \max_{t \in I} r_t. \tag{56}$$

Based on this definition, we derive the connection to our $u_i$ and $v_i$:

1. **Connection to $u_i$:** Every $K$-subset $I \subseteq \mathcal{B}$ with $i \in I$ can be written as $\{i\} \cup S$ where $|S| = K - 1$. Thus:

$$S(i, K, \mathcal{B}) = \frac{\binom{B-1}{K-1}}{\binom{B}{K}} u_i = \frac{K}{B} u_i. \tag{57}$$

2. **Connection to $v_i$:** Consider the sum over the reduced pool $U = \mathcal{B} \setminus \{i\}$:

$$\sum_{j \in U} S(j, K, U) = \frac{1}{\binom{B-1}{K}} \sum_{I \subseteq U, |I|=K} \left( \sum_{j \in I} 1 \right) M(I) = K v_i, \tag{58}$$

since each $K$-subset $I$ contains exactly $K$ indices $j$. Dividing by $B - 1$ yields:

$$\frac{1}{B-1} \sum_{j \neq i} S(j, K, \mathcal{B} \setminus \{i\}) = \frac{K}{B-1} v_i. \tag{59}$$

Substituting these into the definition of Eq. (29):

$$s_i^{loo} = \frac{K}{B} u_i - \frac{K}{B-1} v_i = \gamma \left( u_i - \frac{B}{B-1} v_i \right). \tag{60}$$

Comparing this to our signal $\gamma(u_i - v_i)$, we see a discrepancy in the coefficient of $v_i$. Our method uses $u_i - v_i$, whereas Walder & Karkhanis (2025) implicitly uses $u_i - \frac{B}{B-1}v_i$. Since our EI + L2O baseline $u_i - v_i$ is the unbiased advantage estimator, their $s_i^{\text{sloo}}$ is biased at finite $B$ in that it underestimates the advantage by $\frac{\gamma}{B-1}v_i$. Furthermore, since Walder & Karkhanis (2025) did not run the experiments with the `sloo` in our experiments, we did not include it in our experiments since there is no justification to use it as it is biased. In summary, together with the fact that their `sloo_minus_one` (max@K - max@(K-1)) estimator tends to overestimate the advantage, it becomes clear that both advantage estimators proposed by Walder & Karkhanis (2025) remain biased in opposite directions at finite batch sizes, highlighting the theoretical rigor of our L2O-based approach.

## E. Toy Experiments

In this section, we explain the detailed settings and the additional results of the toy experiments we presented in Sec. 4.

### E.1. Bandits

E.1.1. SETTING

We generate random bandit instances to rigorously evaluate the statistical properties—bias and variance—of the gradient estimators. For each problem instance (seed), we sample the reward vector $\mathbf{r} \in \mathbb{R}^{|\mathcal{A}|}$ and the policy logit vector $\boldsymbol{\theta} \in \mathbb{R}^{|\mathcal{A}|}$ from a standard normal distribution:

$$\boldsymbol{\theta} \sim \mathcal{N}(0, I), \quad \mathbf{r} \sim \mathcal{N}(0, I). \tag{61}$$

The policy is defined as a softmax distribution over actions, $\pi_\theta(a) = \text{softmax}(\boldsymbol{\theta})_a$. Unless otherwise stated, we fix the max@K parameter to $K = 2$ and the batch size to $B = 8$. We construct the PG estimator $\hat{g}$ using EI: $s_i$, EI+L2O: $s_i - b_{-i}^{\text{L2O}}$, and EI+L1O: $s_i - b_{-i}^{\text{L1O}}$. For example, the EI estimator takes the form $\hat{g} = \frac{1}{B} \sum_{i=1}^{B} s_i \nabla_\theta \log \pi_\theta(a_i)$.

**Ground Truth and Estimators.** For a fixed problem instance specified by $(\boldsymbol{\theta}, \mathbf{r})$, we compute the ground-truth max@K policy gradient $g_{\text{true}} := \nabla J^K(\boldsymbol{\theta})$ analytically via the closed-form derivative of the expected improvement (Proposition 1 of Anonymous (2025)). Each gradient estimator $\hat{g}$ is computed from a group of $B$ i.i.d. actions sampled from $\pi_{\boldsymbol{\theta}}$.

**Estimation Error ("Bias") Protocol.** For each fixed instance $(\boldsymbol{\theta}, \mathbf{r})$, we generate $N$ independent gradient estimates $\{\hat{g}_j\}_{j=1}^N$. We measure the (finite-sample) estimation error of the Monte Carlo mean relative to the ground truth:

$$\text{Err}(\boldsymbol{\theta}, \mathbf{r}) := \|\bar{g}_N - g_{\text{true}}\|_2, \qquad \bar{g}_N := \frac{1}{N} \sum_{j=1}^{N} \hat{g}_j. \tag{62}$$

**Empirical Total Variance Protocol.** To quantify estimator stability, we measure the empirical total variance, defined as the trace of the (uncentered) sample covariance:

$$\widehat{\text{TV}}_N(\boldsymbol{\theta}, \mathbf{r}) := \text{Tr}\left( \frac{1}{N} \sum_{j=1}^{N} (\hat{g}_j - \bar{g}_N)(\hat{g}_j - \bar{g}_N)^\top \right) = \frac{1}{N} \sum_{j=1}^{N} \|\hat{g}_j - \bar{g}_N\|_2^2. \tag{63}$$

Equivalently, using the second-moment identity,

$$\widehat{\text{TV}}_N(\boldsymbol{\theta}, \mathbf{r}) = \frac{1}{N} \sum_{j=1}^{N} \|\hat{g}_j\|_2^2 - \|\bar{g}_N\|_2^2. \tag{64}$$

Unless stated otherwise, we use $N = 10^5$ to estimate $\widehat{\text{TV}}_N$.

The total variance of score-function gradients can scale with the action-space dimensionality. To assess variance reduction in high-dimensional regimes (relevant to LLM settings), we report $\widehat{\text{TV}}_N$ across varying numbers of actions.

**Aggregation Across Seeds.** We repeat the above procedure over $L = 10$ independent random seeds, each generating a new pair $(\boldsymbol{\theta}, \mathbf{r})$, and report the mean and standard deviation of $\text{Err}(\boldsymbol{\theta}, \mathbf{r})$ and $\widehat{\text{TV}}_N(\boldsymbol{\theta}, \mathbf{r})$ across seeds.

### E.1.2. ACTION SPACE SIZE AND VARIANCE IN SOFTMAX POLICY

In the bandit setting with a softmax policy, the total variance of the policy gradient estimator is inherently tied to the dimensionality of the action space, denoted by $|\mathcal{A}|$. Here, we derive this relationship analytically.

Consider the score function for a selected action $a$, given by $\nabla_\theta \log \pi_\theta(a) = \mathbf{e}_a - \boldsymbol{\pi}_\theta$, where $\mathbf{e}_a$ is the one-hot vector for action $a$ and $\boldsymbol{\pi}_\theta$ is the probability vector. Assuming the gradient variance is dominated by sampling noise (i.e., neglecting the squared norm of the expected gradient, $\|\mathbb{E}\left[\hat{g}\right]\|^2 \approx 0$), we approximate the total variance by the expected squared norm of the score function:

$$V_{\text{total}}(\boldsymbol{\pi}_\theta) \approx \mathbb{E}_{a\sim\pi_\theta}\left[\|\nabla_\theta \log \pi_\theta(a)\|^2\right]. \tag{65}$$

Expanding the squared norm $\|\mathbf{e}_a - \boldsymbol{\pi}_\theta\|^2$ yields

$$\|\mathbf{e}_a - \boldsymbol{\pi}_\theta\|^2 = (\mathbf{e}_a - \boldsymbol{\pi}_\theta)^\top(\mathbf{e}_a - \boldsymbol{\pi}_\theta) \tag{66}$$
$$= \|\mathbf{e}_a\|^2 - 2\mathbf{e}_a^\top\boldsymbol{\pi}_\theta + \|\boldsymbol{\pi}_\theta\|^2 \tag{67}$$
$$= 1 - 2\pi_\theta(a) + \|\boldsymbol{\pi}_\theta\|^2. \tag{68}$$

Taking expectation over $a \sim \pi_\theta$ gives

$$V_{\text{total}}(\boldsymbol{\pi}_\theta) = \sum_{a=1}^{|\mathcal{A}|} \pi_\theta(a)\left(1 - 2\pi_\theta(a) + \|\boldsymbol{\pi}_\theta\|^2\right) \tag{69}$$
$$= \sum_{a=1}^{|\mathcal{A}|} \pi_\theta(a) - 2\sum_{a=1}^{|\mathcal{A}|} \pi_\theta(a)^2 + \|\boldsymbol{\pi}_\theta\|^2 \sum_{a=1}^{|\mathcal{A}|} \pi_\theta(a) \tag{70}$$
$$= 1 - 2\|\boldsymbol{\pi}_\theta\|^2 + \|\boldsymbol{\pi}_\theta\|^2 \tag{71}$$
$$= 1 - \|\boldsymbol{\pi}_\theta\|^2. \tag{72}$$

Eq. (72) shows that this approximation depends only on the squared $\ell_2$-norm of the probability vector.

To see the effect of the action space size $|\mathcal{A}|$, consider the uniform policy (which is common at initialization), where $\pi_\theta(a) = 1/|\mathcal{A}|$ for all $a$. Then

$$V_{\text{total}}(\boldsymbol{\pi}_{\text{uniform}}) = 1 - \sum_{a=1}^{|\mathcal{A}|}\left(\frac{1}{|\mathcal{A}|}\right)^2 = 1 - |\mathcal{A}| \cdot \frac{1}{|\mathcal{A}|^2} = 1 - \frac{1}{|\mathcal{A}|}. \tag{73}$$

This implies that the variance increases with $|\mathcal{A}|$ and approaches 1 as $|\mathcal{A}| \to \infty$. For example, a binary bandit ($|\mathcal{A}| = 2$) yields $V_{\text{total}}(\boldsymbol{\pi}_{\text{uniform}}) = 0.5$, whereas $|\mathcal{A}| = 1000$ yields $V_{\text{total}}(\boldsymbol{\pi}_{\text{uniform}}) = 0.999$. Thus, large action spaces can exhibit substantially higher gradient variance in the early stages of learning under a softmax policy.

$$V_{\text{total}}(\boldsymbol{\pi}_{\text{uniform}}) = 1 - \sum_{a=1}^{|\mathcal{A}|}\left(\frac{1}{|\mathcal{A}|}\right)^2 = 1 - |\mathcal{A}| \cdot \frac{1}{|\mathcal{A}|^2} = 1 - \frac{1}{|\mathcal{A}|}. \tag{74}$$

This result demonstrates that the variance increases with $|\mathcal{A}|$, asymptotically approaching 1 as $|\mathcal{A}| \to \infty$. For example, a binary bandit ($|\mathcal{A}| = 2$) has an initial variance factor of 0.5, whereas a high-dimensional bandit ($|\mathcal{A}| = 1000$) has a factor of 0.999. Thus, larger action spaces inherently exhibit higher gradient variance in the early stages of learning under a Softmax policy.

### E.1.3. ADDITIONAL RESULTS

Here, we report the additional results for the bandits setting. Especially, we plotted the variance of the gradient estimator with different batch sizes $B \in \{8, 16, 32\}$.

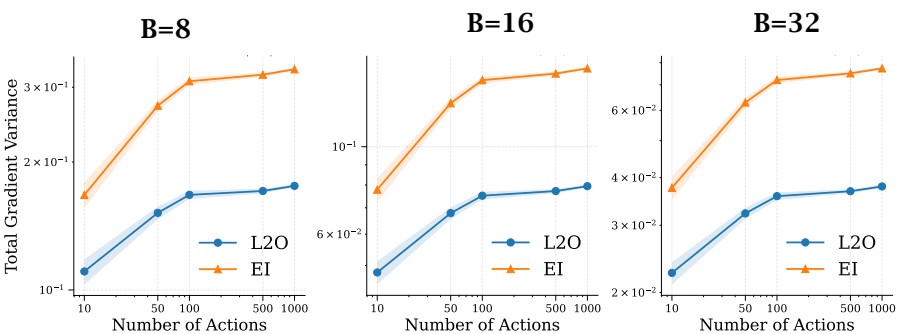

*Figure 7.* Variance vs action space size.

**Variance vs action space size**     Here, we report additional results on variance versus action space size (App. 7). We plot the variance of the gradient estimator for action space sizes $|\mathcal{A}| \in \{10, 50, 100, 1000\}$, with batch sizes $B \in \{8, 16, 32\}$ and $K = 2$. As expected, the variance decreases as the batch size increases. Moreover, the variance-reduction effect of the L2O baseline becomes more pronounced as the action space size increases, consistent with the results in Sec. 4.1.

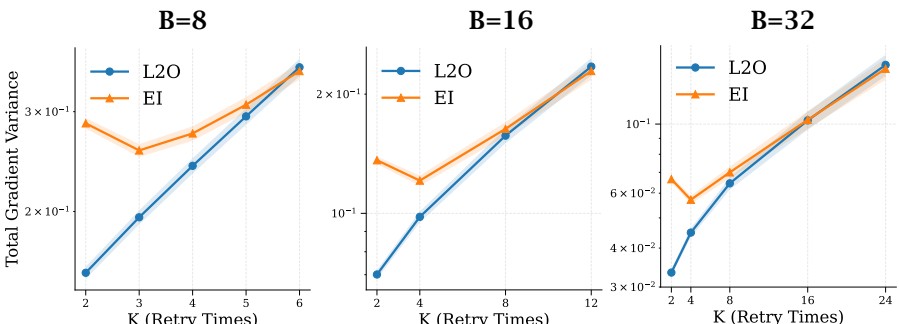

*Figure 8.* Variance vs $K$.

**Variance vs $K$**     Here, we report additional results on variance versus $K$ (App. 8). We plot the variance while varying $K$, with batch sizes $B \in \{8, 16, 32\}$ and $|\mathcal{A}| = 100$. As before, the variance decreases as the batch size increases. We also observe that the variance-reduction effect of the L2O baseline is largest when $K$ is moderate relative to the batch size $B$, which is consistent with the results in Sec. 4.1.

### E.2. Maze Environment

Here, we report the additional results for the maze environment.

#### E.2.1. SETTING

**Estimator Construction for Multi-Step RL**     In episodic RL settings, the optimization objective is defined over the cumulative return of an episode rather than immediate rewards. Let $\tau_i = (s_{i,0}, a_{i,0}, r_{i,0}, \dots)$ denote the $i$-th trajectory in a batch of size $B$. We define the episode return as $R_i = \sum_t r_{i,t}$ and the cumulative score function as $\Psi_i = \sum_t \nabla_\theta \log \pi_\theta(a_{i,t}|s_{i,t})$. To apply the estimators derived in Section 3, we simply substitute the immediate reward $r_i$ with the episode return $R_i$ and the score function $\psi(x, a_i)$ with $\Psi_i$. The specific gradient estimators compared in our experiments are:

- **Standard REINFORCE (with L1O Baseline):** Optimizes the standard expected return $J(\theta) = \mathbb{E}_{x,a\sim\pi_\theta}[R(x,a)]$. We use the standard Leave-One-Out (L1O) baseline to reduce variance:

$$\hat{g}_{\text{std}} := \frac{1}{B}\sum_{i=1}^{B} \Psi_i(R_i - b_{-i}), \quad \text{where } b_{-i} = \frac{1}{B-1}\sum_{j\neq i} R_j. \tag{75}$$

For entropy-regularized REINFORCE, we added the entropy bonus term $\beta H(\pi_\theta)$ to the objective, where $H(\pi_\theta) = -\sum_a \pi_\theta(a) \log \pi_\theta(a)$ and $\beta$ is set to 0.01.

- **EI (Vanilla):** Optimizes the max@K objective using the raw Expected Improvement score $s_i$ (calculated via Eq. (6) in Sec. 3.1 using returns $\{R_j\}_{j=1}^B$):

$$\hat{g}_{\text{ei}} := \frac{1}{B} \sum_{i=1}^B \Psi_i s_i^i. \tag{76}$$

- **EI + L2O (Ours):** Optimizes the max@K objective using the centered advantage $s_i - b_{-i}^{L2O}$:

$$\hat{g}_{\text{l2o}} := \frac{1}{B} \sum_{i=1}^B \Psi_i (s_i - b_{-i}^{L2O}), \tag{77}$$

where $b_{-i}^{L2O}$ is the L2O baseline computed from $\{R_j\}_{j=1}^B$ as defined in Theorem 3.3.

**Hyperparameters.** The learning rate is set to 0.03 after a grid search over $\{0.01, 0.03, 0.05\}$. We collect trajectories with batch sizes $B \in \{5, 8, 16, 32\}$ and train for 3000 iterations. We report the mean and standard error over 10 different random seeds.

**Metrics.** To quantify the variance reduction, we measured the total variance of the gradient estimator $\text{Tr}(\mathbb{V}[\hat{g}])$. In Figure 3 and Figure 9, we report the "Grad Var Ratio", which is defined as the ratio of the gradient variance of our method (EI + L2O) to that of the vanilla EI estimator:

$$\text{Ratio} = \frac{\sum_d \text{Var}(\hat{g}_{\text{l2o}}^{(d)})}{\sum_d \text{Var}(\hat{g}_{\text{ei}}^{(d)})}, \tag{78}$$

where the sum is taken over all parameters. A ratio less than 1.0 indicates effective variance reduction.

E.2.2. ADDITIONAL RESULTS

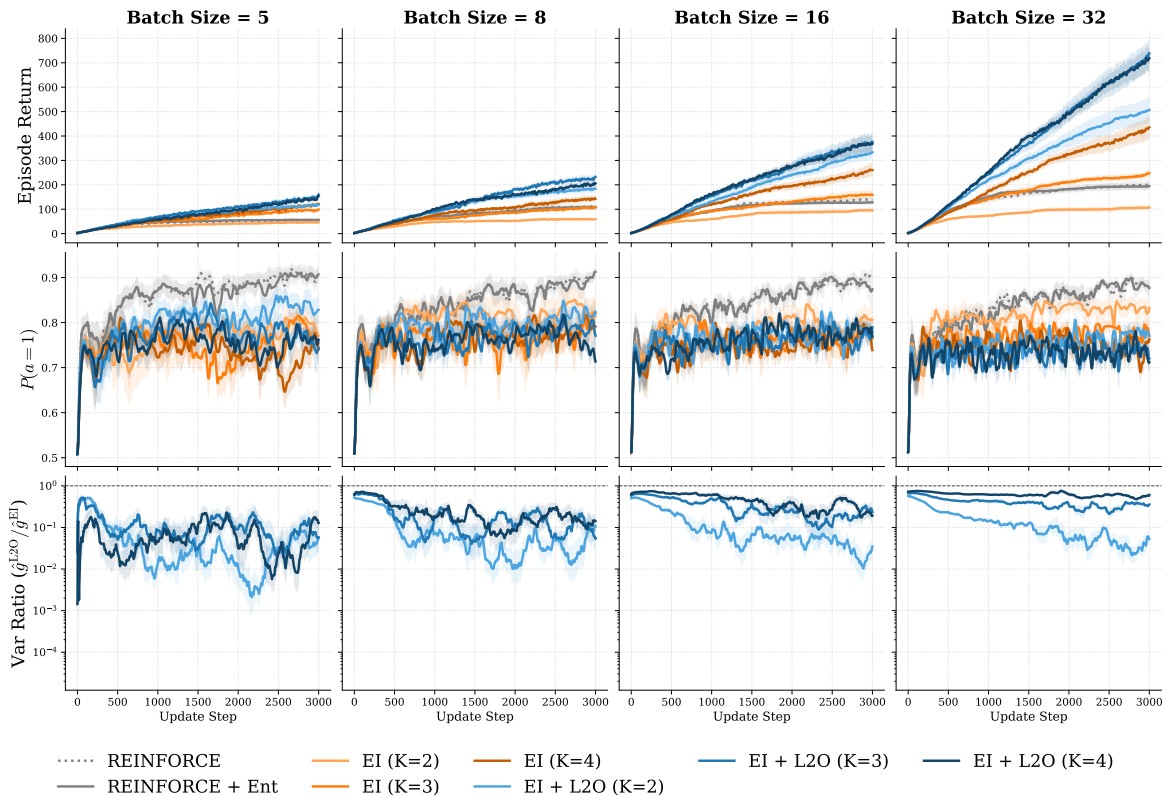

*Figure 9.* Moving average (window size 10) of the average return, probability of action 1 (via global bias $\phi_a$), and gradient variance ratio across different batch sizes $B \in \{5, 8, 16, 32\}$. The error bars are computed using the standard error of the moving average. Our method (EI + L2O) consistently outperforms standard REINFORCE, Entropy-regularized REINFORCE, and vanilla EI. The probability plots indicate that EI-based methods maintain an exploratory policy (prob. $\approx 0.75$) matching the environmental prior, whereas REINFORCE baselines converge to a greedy strategy (prob. $> 0.9$). The gradient variance ratio (EI + L2O / EI) remains consistently below 1.0, demonstrating the effectiveness of the L2O baseline.

In Figure 9, we present the full results for the maze environment across varying batch sizes $B \in \{5, 8, 16, 32\}$, reporting the average return, the probability of selecting action 1 (via the global bias parameter $\phi_a$), and the gradient variance ratio. We observe that **EI + L2O consistently outperforms** all baselines—standard REINFORCE, Entropy-regularized REINFORCE, and vanilla EI—across all tested batch sizes. Aligning with the results in Figure 3, the probability plots reveal that EI + L2O maintains a policy with a strong exploratory tendency (action 1 probability $\approx 0.75$), preventing the collapse to the suboptimal greedy strategy observed in the REINFORCE baselines. Furthermore, the gradient variance ratio is consistently less than 1.0, confirming that the L2O baseline effectively reduces variance in all settings. Notably, the variance reduction effect appears more pronounced when the comparator set size is small (e.g., $K = 2$). This is expected, as increasing the number of comparators ($K$) tends to drive more Expected Improvement values to zero (since it becomes harder to beat the maximum of a larger set), naturally lowering the difference between the EI and that without the L2O baseline. Nevertheless, the L2O baseline provides a consistent benefit across all configurations.

## F. LLM Experiments

In this section, we provide additional results for the LLM experiments presented in Sec. 5.

### F.1. SETTING

All models are trained with a learning rate of $10^{-6}$, a batch size of 1024, and a optimization mini-batch size of 256. For each input problem, we roll out 8 responses using a temperature of 1.0. Qwen model is trained on the MATH dataset (Hendrycks

*Table 2.* **Task-average pass@k** ($k \leq 256$). Unweighted average over AIME24, AIME25, AMC23, MATH500, and Minerva under the evaluation protocol in Sec. 5.1.

| Method | 1 | 2 | 4 | 8 | 16 | 32 | 64 | 128 | 256 |
|---|---|---|---|---|---|---|---|---|---|
| *Qwen2.5-Math-7B* | | | | | | | | | |
| GRPO | **40.7** | 45.5 | 49.2 | 52.2 | 55.4 | 58.7 | 62.1 | 65.6 | 69.3 |
| Entropy-Adv | 38.4 | 43.7 | 48.3 | 52.3 | 56.3 | 60.2 | 64.1 | 67.7 | 70.9 |
| PKPO | 37.2 | 42.6 | 47.4 | 51.4 | 55.1 | 59.0 | 62.9 | 66.8 | 70.6 |
| Ours | 39.9 | **46.5** | **51.7** | **56.2** | **60.1** | **63.8** | **67.5** | **71.1** | **74.3** |
| *Llama-3.2-3B-Instruct* | | | | | | | | | |
| GRPO | **22.4** | **28.0** | 33.3 | 38.2 | 42.9 | 47.7 | 52.7 | 57.5 | 62.1 |
| Entropy-Adv | 22.1 | 27.7 | 33.2 | 38.5 | 43.6 | 48.5 | 53.3 | 58.2 | 63.1 |
| PKPO | 20.1 | 25.5 | 31.3 | 37.1 | 42.5 | 47.5 | 52.7 | 58.0 | 63.5 |
| Ours | 21.8 | 27.6 | **33.4** | **38.7** | **43.8** | **48.9** | **54.0** | **59.5** | **65.0** |

*Table 3.* **Per-benchmark pass@1 / pass@256.** Results for each benchmark under the same evaluation protocol as Sec. 5.1.

| Method | AIME24 | AIME25 | AMC23 | MATH500 | Minerva | Avg. |
|---|---|---|---|---|---|---|
| *Qwen2.5-Math-7B* | | | | | | |
| GRPO | **30.3**/72.0 | 10.5/41.1 | **63.3**/96.1 | **76.3**/92.9 | **23.0**/44.5 | **40.7**/69.3 |
| Entropy-Adv | 25.6/72.3 | 9.8/47.3 | 59.5/97.0 | 74.7/93.6 | 22.4/44.2 | 38.4/70.9 |
| PKPO | 19.9/72.9 | **11.1**/49.2 | 60.7/94.4 | 72.9/93.1 | 21.4/43.4 | 37.2/70.6 |
| Ours | 28.5/**74.2** | 10.6/**54.7** | 62.2/**98.0** | 75.6/**95.7** | 22.9/**48.8** | 39.9/**74.3** |
| *Llama-3.2-3B-Instruct* | | | | | | |
| GRPO | **13.7**/50.5 | 0.5/34.3 | **30.2**/93.5 | 52.0/90.5 | **15.5**/41.6 | **22.4**/62.1 |
| Entropy-Adv | 12.6/49.2 | **0.9**/36.5 | 29.5/94.8 | **52.4**/91.7 | 15.1/43.2 | 22.1/63.1 |
| PKPO | 9.3/54.4 | 0.5/34.0 | 27.4/**96.8** | 49.2/90.7 | 14.2/41.4 | 20.1/63.5 |
| Ours | 11.7/**58.1** | 0.6/35.4 | 29.9/96.0 | 51.7/**91.8** | 15.3/**43.8** | 21.8/**65.0** |

et al., 2021), while the Llama model is trained on a combined dataset of GSM8K (Cobbe et al., 2021) and MATH (Level 1).

## F.2. Additional Results

Tables 2 and 3 summarize the same evaluation protocol as the main text, reporting (i) task-average pass@k up to $k \leq 256$ and (ii) per-benchmark pass@1 and pass@256. Figure 10 provides the per-benchmark pass@k curves to visualize how improvements distribute across datasets and across inference compute.

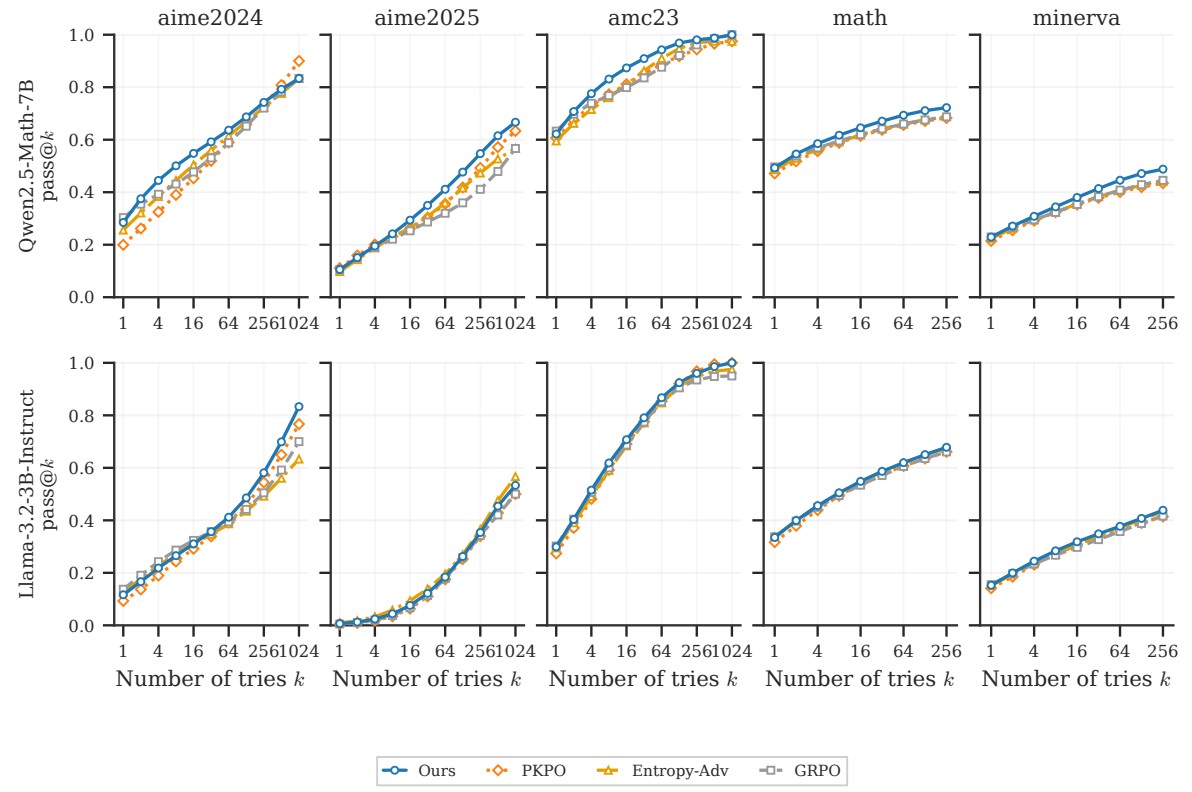

*Figure 10.* **Per-benchmark pass@k curves.** For each benchmark, we plot pass@k as a function of inference compute $k$ for all methods. This complements the task-average curves in Fig. 4 by showing where gains come from across datasets and whether improvements persist as $k$ increases.

## G. LLM Usage

LLMs were used for editing the manuscript, and as an algebra tool to help with derivations based on suggestions by the authors. The key research ideas were proposed by the authors.

