# OpenReview forum: "Pass@K Policy Optimization with Unbiased Advantage Estimation"
_ICML.cc/2026/Conference — Submitted to ICML 2026_

### Official Review · Reviewer_jkfT · 2026-02-20

**Soundness:** 2
**Presentation:** 3
**Significance:** 3
**Originality:** 3
**Overall Recommendation:** 4
**Confidence:** 3

**Summary:**

This paper studies the policy gradient of learning pass@K objective, with a special focus on variance reduction. The paper first identify the two key feature of baseline, unbiased estimation of policy gradient and zero-expectation for variance minimization. To comply with this two requests, the paper proposed a baseline design for pass@K policy optimization, which is the average of scalar term $s$ computed over the reduced subset in the batch. The paper further proposes an $O(B^2)$ method to compute such baseline. Finally, the paper presents experiment results on both toy setting and LLM setting.

**Compliance With Llm Reviewing Policy:**

Affirmed.

**Final Justification:**

This paper studies the problem of pass@K objective. In particular, this paper introduces a variance-reduced and unbiased policy gradient for pass@K objective, which is the first of its kind and of considerable signficance. The empirical verification also indicates that the proposed method improves over previous baselines.

However, overall, the method is confined to training with $K=2$. While the authors clarifies in the rebuttal that this restriction is primarily due to pass@K objective, it somewhat narrows the scope of this paper. Therefore, I maintain my evaluation to weak accept.

**Key Questions For Authors:**

N/A

**Limitations:**

Yes

**Strengths And Weaknesses:**

The strengths and weaknesses of this paper are listed as follows

Strengths:

1. This paper proposes a variance-reduced and unbiased policy gradient for pass@K objective, which is new to the reviewer and also practical relevant.

2. This paper further provides an efficient method for computing the proposed baseline, making it implementable.

2. The paper is in general clearly written and well-organized.

Weaknesses:

1. The experiment is limited to Llama-3.2-3B-Instruct and Qwen2.5-Math-7B. To my knowledge, these models are relatively weak compared to new SOTA. Could the author alsp provide results on stronger models?

2. Also in LLM reasoning experiments, the training objective is restricted to $K=2$ and $K=4$, lacking an study with larger $K$.

3. While $O(B^2)$ is a significant improvment over naive $O(B^3)$, it might still incur a computational burden especially when batch size is large. Could the author also report the computational time overhead introduced by the proposed method (compared to baselines like EI, EI+L1O).

---

> ### Author Rebuttal · Authors · 2026-03-31
>
> Thank you for the review.
>
> >The experiment is limited to Llama-3.2-3B-Instruct and Qwen2.5-Math-7B. To my knowledge, these models are relatively weak compared to new SOTA. Could the author alsp provide results on stronger models?
>
> To address the concern, we ran a comparison of PKPO against Ours (training with $K=2$) on Qwen3-4B-Instruct and the results are in Figure 6 in https://anonymous.4open.science/r/remax-icml-rebuttal/README.md
> The results show significant improvements compared to PKPO across all evaluation $k$ values. Due to resource and time constraints, we have not been able to include GRPO results, but will do so in a month or so.
>
> ### Qwen3-4B-Instruct results
>
> | Method | k=1 | k=2 | k=4 | k=8 | k=16 | k=32 | k=64 | k=128 | k=256 |
> |---|---:|---:|---:|---:|---:|---:|---:|---:|---:|
> | Ours | **59.9%** | **65.0%** | **68.5%** | **71.0%** | **72.8%** | **74.5%** | **76.3%** | **77.7%** | **78.9%** |
> | PKPO | 57.5% | 62.6% | 66.5% | 69.5% | 72.0% | 74.1% | 75.7% | 76.9% | 77.8% |
>
> >Also in LLM reasoning experiments, the training objective is restricted to $K=2$ and $K=4$, lacking an study with larger $K$.
>
> To address this concern, we also ran experiments with $K=6$ (see figure 6 in https://anonymous.4open.science/r/remax-icml-rebuttal/README.md). We see that performance drops compared to $K\in\{2, 4\}$. Note that the magnitude of $K$ is limited by the group size, which is set to 8. So we have already tested a wide range of $K$ relative to the the size of the group. In our experience (in the supplementary attached anonymous paper), using too large a $K$ tends to reduce performance, and we recommend smaller values like 2 or 4. Below we also give the results as a table.
>
> ### Qwen2.5-Math-7B K Sensitivity
>
> | Method | k=1 | k=2 | k=4 | k=8 | k=16 | k=32 | k=64 | k=128 | k=256 |
> |---|---:|---:|---:|---:|---:|---:|---:|---:|---:|
> | Base | 25.8% | 35.5% | 44.0% | 50.7% | 55.7% | 59.7% | 63.4% | 67.4% | 71.6% |
> | GRPO | **40.7%** | 45.5% | 49.2% | 52.2% | 55.4% | 58.7% | 62.1% | 65.6% | 69.3% |
> | PKPO | 37.2% | 42.6% | 47.4% | 51.4% | 55.1% | 59.0% | 62.9% | 66.8% | 70.6% |
> | Ours (K=2) | 39.9% | **46.5%** | **51.7%** | **56.2%** | **60.1%** | **63.8%** | 67.5% | 71.1% | 74.3% |
> | Ours (K=4) | 36.6% | 44.3% | 50.2% | 55.2% | 59.6% | 63.7% | **67.6%** | **71.5%** | **75.4%** |
> | Ours (K=6) | 34.7% | 42.8% | 49.1% | 54.2% | 58.5% | 62.4% | 66.0% | 69.3% | 72.5% |
>
> >While $O(B^2)$ is a significant improvment over naive $O(B^3)$, it might still incur a computational burden especially when batch size is large. Could the author also report the computational time overhead introduced by the proposed method (compared to baselines like EI, EI+L1O).
>
> The computation costs from the logs during training are below (in ms).
>
> | Metric | Time (ms) |
> |---|---:|
> | timing_s/gen | 152.534 |
> | timing_s/old_log_prob | 39.662 |
> | timing_s/reward_fn | 83.900 |
> | timing_s/baseline | **0.134** |
> | timing_s/adv | 84.349 |
> | timing_s/update_actor | 156.895 |
> | timing_s/step | 433.784 |
>
> We see that the cost spent on baseline computation is miniscule compared to total computation time. Additionally, we ran a computation time benchmark for larger batch sizes and compared against the implementation from Walder et al. (2025, in their appendix), and the results are in the final figure in https://anonymous.4open.science/r/remax-icml-rebuttal/README.md. We see that our implementation is fast and do not expect the baseline computation time to add any noticeable overhead in practical scenarios (even with batch sizes of ~1000 the baseline computation takes on the order of **1ms**). We will add more clarification about this in the paper.

---

> > ### Author Rebuttal · Reviewer_jkfT · 2026-03-31
> >
> > I thank the author for the response and experiments. My remaining questions are listed as follows
> >
> > 1. Could the author explain why $K$ is limited to 8 at training? Is it due to batch size (i.e., GPU memory) limitation, the limitation of pass@K training objective or your proposed method?
> >
> > 2. Could the author provide some brief explanation on why using large $K$ during training reduces the preformance?

---

> > > ### Author Response · Authors · 2026-04-02
> > >
> > > Thank you for the clarifying questions.
> > >
> > > 1. We would like to clarify some terminology. The batch size is 1024, the group size is $G=8$ and the minibatch size is 256. These are default values in the verl code that we used as a basis for our experiments. What this means is that the method will sample 1024 prompts, and for each prompt it generates $G=8$ outputs. Then it updates the LLM using minibatches of size 256 with 8 outputs per prompt. These are standard values in such LLM post-training experiments, but sometimes $G=16$ is also used. Typically on the small end one would use $G=4$, and on the higher end we have seen $G=64$ as well. As the baseline is computed within each group, the magnitude of the retry parameter $K$ is limited by the group size $G$. Moreover, to use the L2O baseline in practice $K$ is limited to below $G-1$. So, to answer your question, the setting of $G=8$ at training comes from the implementations that we use, and it is set to this value for efficiency of learning. This is not due to any limitation, but simply because it performs better than large group sizes (generating too many outputs per a single prompt can be wasteful instead of generating outputs for more prompts).
> > >
> > > 2. We believe the issue is due to a reduction in the training signal. When $K$ becomes large, out of every set of $K$ samples, only the largest one is used as a reward, so the number of used samples and information reduces and it may not be as efficient. Another point is that if $K$ is large then even if many generated samples are incorrect, as long as there is one correct sample the generation is regarded a success. So once the LLM produces a few correct samples for a prompt, the training signal for that prompt disappears as the metric does not improve even if more correct samples are generated. We believe this may reduce efficiency. This may perhaps be overcome by training for longer. In the results we showed you earlier, $K=4$ worked better than $K=2$ for large values of $k$ in the $pass@k$ metric based evaluation (while at the same time, both are improving over the base model), so it is plausible that with larger $K$ and more training time, the performance on large $k$ in the metric may further improve.

---

### Official Review · Reviewer_U8Pq · 2026-03-06

**Soundness:** 2
**Presentation:** 2
**Significance:** 2
**Originality:** 2
**Overall Recommendation:** 3
**Confidence:** 2

**Summary:**

This paper introduces a method for optimizing the pass@K (or max@K) metric in RL. The authors identify that existing policy gradient estimators for this objective yield biased advantage estimates that are not zero-mean. To address this, they propose the Leave-Two-Out (L2O) baseline, which constructs a centered advantage term. They further develop a vectorized algorithm to compute the L2O baseline in O(B²) time. Empirical validation is conducted on some toy experiments and LLM reasoning tasks.

**Compliance With Llm Reviewing Policy:**

Affirmed.

**Key Questions For Authors:**

See above.

**Limitations:**

See above.

**Strengths And Weaknesses:**

**Strengths**

1. The paper is generally well-written and easy to follow.
2. The authors provide a vectorized algorithm to compute the L2O baseline in O(B²) time, making it feasible for large-scale LLM training where the batch size B is moderate.
3. The method is designed as a simple reward transformation, allowing seamless integration into existing group-based RL frameworks like GRPO.


**Weaknesses**

1. While reduced to O(B²), the L2O computation is inherently more complex than standard policy gradient methods that use simpler batch-mean baselines.
2. The evaluation is primarily focused on math reasoning tasks with verifiable rewards; its effectiveness on more subjective tasks (e.g., open-ended dialogue) that lack binary feedback is not explored.
3. The proposed method shows performance degradation at Pass@1. As shown in Table 2 and Table 3 in the Appendix, it consistently underperforms standard GRPO on this metric, raising questions about the practical necessity of the more complex L2O method for optimizing Pass@1 performance.
4. Can the authors explain why their method underperforms GRPO on the Pass@1 metric? In LLM post-training (RL), we usually aim to improve the model's Pass@1 performance.

---

> ### Author Rebuttal · Authors · 2026-03-31
>
> Thank you for the review.
>
> >While reduced to O(B²), the L2O computation is inherently more complex than standard policy gradient methods that use simpler batch-mean baselines.
>
> Yes, however it is still simple. We will provide code, and using it is just 1 line change in the code, where the user has to transform their rewards: “new_reward_batch = ei_l2o(old_reward_batch)”.
> Everything else about the users code can remain unchanged.
>
> Regarding computational overhead, it is miniscule. Below are the logged computation times for different parts of the computation in 1 step (in ms):
>
> | Metric | Time (ms) |
> |---|---:|
> | timing_s/gen | 152.534 |
> | timing_s/old_log_prob | 39.662 |
> | timing_s/reward_fn | 83.900 |
> | timing_s/baseline | **0.134** |
> | timing_s/adv | 84.349 |
> | timing_s/update_actor | 156.895 |
> | timing_s/step | 433.784 |
>
> We see that the computation time spent on the baseline computation is extremely small compared to other parts of the computation. We also added a computation time benchmark (https://anonymous.4open.science/r/remax-icml-rebuttal/README.md) for larger batch sizes, and even for batch sizes like 1000, the computation time is on the order of 1ms, which is miniscule. We appreciate that this may not be completely clear from the paper, and will add the results in the paper.
>
> >The evaluation is primarily focused on math reasoning tasks with verifiable rewards; its effectiveness on more subjective tasks (e.g., open-ended dialogue) that lack binary feedback is not explored.
>
> Our experiments intentionally focus on verifier-based RL, which is the standard GRPO/RLVR regime and lets us isolate the effect of the estimator without confounding reward-model noise. Preference-based dialogue or summarization is an important extension, but it introduces an additional learned judge/reward model whose noise and bias are orthogonal to the baseline-centering issue studied here. Additionally, the method is applicable to non-binary rewards, e.g., the toy maze experiments that we run.
>
> >The proposed method shows performance degradation at Pass@1. As shown in Table 2 and Table 3 in the Appendix, it consistently underperforms standard GRPO on this metric, raising questions about the practical necessity of the more complex L2O method for optimizing Pass@1 performance.
>
> For reference, we show the Avg result column from table 2 below:
>
> | Model | Method | pass@1 (Avg.) | pass@256 (Avg.) |
> |---|---|---:|---:|
> | Qwen2.5-Math-7B | GRPO | **40.7** | 69.3 |
> | Qwen2.5-Math-7B | PKPO | 37.2 | 70.6 |
> | Qwen2.5-Math-7B | Ours | 39.9 | **74.3** |
> | Llama-3.2-3B-Instruct | GRPO | **22.4** | 62.1 |
> | Llama-3.2-3B-Instruct | PKPO | 20.1 | 63.5 |
> | Llama-3.2-3B-Instruct | Ours | 21.8 | **65.0** |
>
>
> A common issue with methods like GRPO is that even though they increase pass@1, they end up reducing pass@k compared to the base model. This is why methods for pass@k policy optimization such as PKPO (our main competing method) were developed. The results show that compared to PKPO, the drop in pass@1 is smaller, while simultaneously, the increase in pass@k is much larger. Moreover, the overall pass@1 reduction compared to GRPO is not that large. To further justify the significance of the empirical gains compared to PKPO (Walder et al. NeurIPS2025 spotlight), we added experiments comparing the pass@k metric against the base model (see figure 5 at https://anonymous.4open.science/r/remax-icml-rebuttal/README.md). Note that the closest competitor Walder et al. had not included such comparisons with the base model in their paper. We see from our results that competing methods including PKPO do not improve the pass@k metric on these experiments relative to the base model. Our method, on the other hand, does improve compared to the base model (the improvement on the 7B model on reasoning tasks is particularly relevant). We believe this is a conceptual improvement in the performance. Moreover, the improvement size is significant in terms of LLM performance. In the link, we also added an analysis of shrinkage/expansion (figure 12 in the link) and our method expands the set of problems with at least 1 correct solution from many generations.
>
> >Can the authors explain why their method underperforms GRPO on the Pass@1 metric? In LLM post-training (RL), we usually aim to improve the model's Pass@1 performance.
>
> This is common for pass@k optimization methods (e.g., the main competing method PKPO also has this issue, and to a greater extent than ours), and we expected it. See the previous response above for more discussion on this. Also note that PKPO (Walder et al. 2025 NeurIPS spotlight) also proposed to anneal the K parameter in the optimization, and showed that by annealing they can maintain the pass@1 performance. We could do the same, but the drop in pass@1 performance is already much smaller than that of PKPO. We will add a note about this in the paper.

---

> > ### Author Rebuttal · Reviewer_U8Pq · 2026-04-03
> >
> > Given that the method is limited to verifiable rewards and only improves performance on Pass@256 at the cost of a drop in Pass@1, I do not see much practical value in real applications. Therefore, I have decided to keep my original score.

---

> > > ### Author Response · Authors · 2026-04-04
> > >
> > > Thank you for the follow-up. Three points are central here: (1) the improvement is already visible at pass@2, not only at pass@256; (2) pass@k/max@k are practically relevant when the system can generate a small number of candidates and then verify or rerank them; and (3) while our experiments focus on verifiable rewards, the method itself is defined for general scalar rewards.
> > >
> > > First, the gain is not only at pass@256. On Qwen2.5-Math-7B, our K=2 setting already improves over GRPO at **pass@2**. In our earlier reply we only showed a small excerpt of the appendix results, which may have created the impression that the benefit appears only at very large k. To make this easier to verify, we provide the per-k results for the most relevant baselines below, now also including the base model. (To avoid ambiguity: in the table below, K in “Ours (K=2/4)” is the training objective size, while k in the columns is the evaluation / inference budget.)
> > >
> > > ### Qwen2.5-Math-7B Full Results with Comparison to Base
> > >
> > > | Method | k=1 | k=2 | k=4 | k=8 | k=16 | k=32 | k=64 | k=128 | k=256 |
> > > |---|---:|---:|---:|---:|---:|---:|---:|---:|---:|
> > > | Base | 25.8% | 35.5% | 44.0% | 50.7% | 55.7% | 59.7% | 63.4% | 67.4% | 71.6% |
> > > | GRPO | **40.7%** | 45.5% | 49.2% | 52.2% | 55.4% | 58.7% | 62.1% | 65.6% | 69.3% |
> > > | PKPO | 37.2% | 42.6% | 47.4% | 51.4% | 55.1% | 59.0% | 62.9% | 66.8% | 70.6% |
> > > | Ours (K=2) | 39.9% | **46.5%** | **51.7%** | **56.2%** | **60.1%** | **63.8%** | 67.5% | 71.1% | 74.3% |
> > > | Ours (K=4) | 36.6% | 44.3% | 50.2% | 55.2% | 59.6% | 63.7% | **67.6%** | **71.5%** | **75.4%** |
> > >
> > >
> > > A further **important observation** from this table is that, among the methods shown here, GRPO and PKPO eventually fall below the base model at larger k ($k \ge 16$), whereas our method remains above the base model across the full k range in this experiment. So the benefit is not only at k=256, and it is not only relative to other post-training baselines. This also explains the pass@1 trade-off: as shown by our K-ablation in the paper, training for larger-K objectives broadens exploration across multiple valid solutions, which can slightly reduce the top-1 mode while improving the probability that at least one of several samples is correct.
> > >
> > > Second, inference-time objectives such as pass@k and max@k are practically relevant when an application can generate multiple candidates and then select one using a verifier (pass@k) or a learned judge / reward model (max@k / Best-of-N). Some settings are naturally pass@k---for example, in code generation it is often enough that one of a small number of sampled programs passes the test suite. This is precisely the setting studied in recent work on pass@k/max@k and Best-of-N decoding: Tang et al. (2025), Walder & Karkhanis (2025), Chen et al. (2025b), Peng et al. (2025), and Bagirov et al. (2025) explicitly optimize inference-time objectives such as pass@k/max@k, while Ichihara et al. (2025) and Sun et al. (2024) study Best-of-N decoding as an effective inference-time strategy. OpenAI also discusses best-of-n / reranking as a practical and competitive approach (WebGPT; Measuring Goodhart’s Law). Therefore, when a system can sample even a small number of candidates and then verify or rank them, the relevant deployment objective is not only pass@1 but also pass@k/max@k. This is why improvement already at k=2 is practically meaningful, not only improvement at k=256. We will clarify this framing in the introduction and expand Appendix A with a short discussion of inference-time strategies and related references.
> > >
> > > Finally, while our experiments focus on verifier-based reasoning tasks, the method itself is not limited to verifiable rewards. Methodologically, L2O is formulated for max@K, the continuous-reward generalization of pass@K. We chose verifier-based benchmarks because they are the most standard setting in this literature and let us isolate the estimator issue without confounding from learned-judge noise or reward hacking. The toy maze experiments already go beyond binary final-answer correctness. We agree, however, that additional validation on subjective tasks with learned judges would strengthen the paper, and we will state this limitation more clearly.
> > >
> > > We hope this clarifies both the practical regime targeted by our method and the intended scope of the empirical evaluation.
> > >
> > > References
> > >
> > > Tang et al. (2025): https://arxiv.org/abs/2503.19595
> > >
> > > Walder & Karkhanis (2025): https://arxiv.org/abs/2505.15201
> > >
> > > Ichihara et al. (2025): https://arxiv.org/abs/2502.12668
> > >
> > > Sun et al. (2024): https://arxiv.org/abs/2410.20290
> > >
> > > OpenAI (2022), Measuring Goodhart’s law: https://openai.com/index/measuring-goodharts-law/
> > >
> > > OpenAI (2021), WebGPT: https://openai.com/index/webgpt/

---

### Official Review · Reviewer_NHfd · 2026-03-13

**Soundness:** 3
**Presentation:** 3
**Significance:** 3
**Originality:** 3
**Overall Recommendation:** 5
**Confidence:** 4

**Summary:**

This work presents a novel Expected Improvement perspective on pass@K policy optimisation, and uses this perspective to derive a practical algorithm that reduces policy gradient variance. They provide a theoretical and empirical analysis, showing that on both toy and real-world tasks their algorithm leads to significantly reduced policy variance and improved pass@K returns over prior methods.

**Compliance With Llm Reviewing Policy:**

Affirmed.

**Key Questions For Authors:**

The identity on line 153 appears to have an error: LHS is missing a max; also, line 157 does not make sense to me. Is the inner max_k intentional? the k is interfering with the dummy k in the outer term.

**Limitations:**

The authors have not included any discussion of the limitations.

**Strengths And Weaknesses:**

Strengths:
- elegant conceptual perspective: the expected improvement connection is interesting and opens up interesting follow-up perspectives in connection to other Bayesian optimisation related concepts (e.g. perhaps there is a connection to be made with knowledge gradient or other EI-adjacent algorithms)
- practical algorithm, the vectorisation approach is sensible
- broad relevance, variance reduction for pass@K is a broadly applicable contemporary issue, and the authors motivate their case well with relevant literature
- strong theoretical analysis
- well written

Weaknesses:
- real-world experimental results show marginal improvement
- no discussion of limitations of their work

---

> ### Author Rebuttal · Authors · 2026-03-31
>
> Thank you for the review.
>
> >real-world experimental results show marginal improvement
>
> To further justify the significance of the empirical gains compared to EI (PKPO, Walder et al. NeurIPS2025 spotlight), we added experiments comparing the pass@k metric against the base model (see figure at https://anonymous.4open.science/r/remax-icml-rebuttal/README.md). Note that the closest competitor Walder et al. had not included such comparisons with the base model in their paper. We see from our results that competing methods including PKPO do not improve the pass@k metric on these experiments relative to the base model. Our method, on the other hand, does improve compared to the base model (the improvement on the 7B model on reasoning tasks is particularly relevant). We believe this is a conceptual improvement in the performance thanks to our method. Moreover, the improvement size is significant in terms of LLM performance. We give the results in the link as a table below as well.
>
> In the link, we also added an analysis of shrinkage/expansion (figure 12 in the link) and our method expands the set of problems with at least 1 correct solution from many generations.
>
> ### Qwen2.5-Math-7B Results Including the Base Model
>
> | Method | k=1 | k=2 | k=4 | k=8 | k=16 | k=32 | k=64 | k=128 | k=256 |
> |---|---:|---:|---:|---:|---:|---:|---:|---:|---:|
> | Base | 25.8% | 35.5% | 44.0% | 50.7% | 55.7% | 59.7% | 63.4% | 67.4% | 71.6% |
> | GRPO | **40.7%** | 45.5% | 49.2% | 52.2% | 55.4% | 58.7% | 62.1% | 65.6% | 69.3% |
> | PKPO | 37.2% | 42.6% | 47.4% | 51.4% | 55.1% | 59.0% | 62.9% | 66.8% | 70.6% |
> | Ours (K=2) | 39.9% | **46.5%** | **51.7%** | **56.2%** | **60.1%** | **63.8%** | 67.5% | 71.1% | 74.3% |
> | Ours (K=4) | 36.6% | 44.3% | 50.2% | 55.2% | 59.6% | 63.7% | **67.6%** | **71.5%** | **75.4%** |
>
> >no discussion of limitations of their work
>
> One possible limitation of pass@k optimization in general (not limited to our method) is that the pass@1 performance is slightly less than the regular GRPO. Walder’s work (PKPO) also mentioned this, and noted that they can anneal the $K$ parameter and maintain the pass@1 performance while still increasing pass@k. In our case the drop in pass@1 is not as large as that of PKPO, and already tiny compared to the increase in pass@k, so it is not a big problem, but we could also add an annealing stage at the end of training. We will add a sentence about this in the conclusions section. In general, our paper was about variance reduction, and we believe the method clearly achieves this, and the evidence is substantial.
>
> >The identity on line 153 appears to have an error: LHS is missing a max; also, line 157 does not make sense to me. Is the inner max_k intentional? the k is interfering with the dummy k in the outer term.
>
> There is no error, but it was written in a confusing way due to both having a $k$ and a $-k$, and the two not being the same. The way to parse the equation is $(\max_i r_i) - W_{-k} = (r_k - W_{-k})_+ $, where the plus is the ReLU.
>
> The reason this equation holds is that $W_{-k}$ is the max among all indexes $i \neq k$, so when an $r_i > r_k$, then the subtraction just becomes $r_i - r_i=0$. The only non-zero case is when $r_k$ is the largest among the $K$ rewards (then it becomes $r_k - r_i$, where $r_i$ is the largest among the other ones, denoted as $W_{-k}$). We will improve the clarity of this explanation; thank you for pointing it out.

---

> > ### Author Rebuttal · Reviewer_NHfd · 2026-04-06
> >
> > The authors have thoughtfully engaged with my concerns and questions. I appreciate their responses and keep my high score for this paper.

---

### Official Review · Reviewer_sKfN · 2026-03-16

**Soundness:** 3
**Presentation:** 3
**Significance:** 2
**Originality:** 3
**Overall Recommendation:** 4
**Confidence:** 3

**Summary:**

For complex reasoning tasks, Reinforcement Learning (RL) has become a typical approach for post-training Large Language Models (LLMs).
In such settings, effective exploration is crucial, and prior works have employed classical RL techniques (e.g., entropy regularization and count-based exploration) to encourage exploration.
However, these approaches require hard to balancing between auxiliary exploration objective with the primary task reward.

As an alternative, recent works propose directly optimizing pass@K (or its generalization, max@K), which incentivizes answer diversity by maximizing the probability of obtaining at least one correct answer in K independent samples.
While existing pass@K policy optimization algorithms propose methods to reduce variance of the estimator, this paper revisits pass@K policy gradient and introduces Leave-Two-Out (L2O) baselines to further reduce the variance of pass@K policy gradient estimator.
The authors additionally introduce a vectorized algorithm that enables efficient computation of the L2O baseline.

The authors empirically show that the proposed method reduces variance in bandit settings and improves performance in the Maze environment compared to Expected Improvement (EI).
For LLM reasoning tasks, the authors demonstrate consistent improvements over several baselines, including PKPO and GRPO.

**Compliance With Llm Reviewing Policy:**

Affirmed.

**Final Justification:**

I remain inclined toward acceptance.

**Key Questions For Authors:**

- The main motivation of this paper is reducing gradient variance. While gradient variance is successfully reduced, the improvement in downstream reasoning performance appears relatively modest. Could the authors provide further analysis on the relationship between variance reduction and final performance?
- Under what conditions does the proposed L2O baseline provide the largest benefits. When does L2O help the most? (in terms of group size B, objective size K, reward sparsity, or task difficulty, etc.).
- Additional comparison with other baselines should be provided. Alternatively, approaches such as Best-of-N optimization or search-based methods may be complementary rather than strictly competing. Could the authors discuss whether L2O can be combined with such methods or how it would behave in those settings?

**Minor comments**:
- In line 135, It seems that the citation (Anonymous, 2022) is not a meaningful citation.
- Typo: In line 59 (page 2): missing dot after (Williams, 1992)

Overall, I enjoyed the derivations and intuitions behind L2O, but it remains unclear whether the proposed method is truly effective in practice. Thus, while I assigned a weak accept (4), I am really on the borderline (3.5).

**Strengths And Weaknesses:**

**Strengths**
- This paper provides a clear and well-organized theoretical analysis of pass@K policy gradient estimators and highlights that existing EI-based estimators leave room for further variance reduction.
- The proposed L2O baseline is simple and principled (i.e., requiring no additional heuristics or hyperparameter tuning) and can be easily integrated into existing RL algorithms.
- The vectorized formulation enables efficient computation in large-batch settings.
- Easy to follow

**Weaknesses**
- While the proposed method consistently improves performance over EI, the empirical gains appear relatively modest, especially for reasoning tasks.
- The set of baseline approaches is somewhat limited (e.g., Best-of-N or MCTS-based methods).

---

> ### Author Rebuttal · Authors · 2026-03-31
>
> Thank you for the review.
>
> >While the proposed method consistently improves performance over EI, the empirical gains appear relatively modest, especially for reasoning tasks.
>
> To further justify the significance of empirical gains compared to EI (PKPO, Walder et al. NeurIPS2025 spotlight), we added experiments comparing the pass@k metric against the base model (see figure 5 at https://anonymous.4open.science/r/remax-icml-rebuttal/README.md). Note that the closest competitor Walder et al. had not included such comparisons with the base model in their paper. We see that competing methods including PKPO do not improve the pass@k metric on these experiments relative to the base model. Our method, on the other hand, does improve compared to the base model (the improvement on the 7B model on reasoning tasks is particularly relevant). We believe this is a conceptual improvement in performance. Moreover, the improvement size is significant in terms of LLM performance. In the link, we also added an analysis of shrinkage/expansion (figure 12 in the link) and our method expands the set of problems with at least 1 correct solution from many generations.
>
> >The set of baseline approaches is somewhat limited (e.g., Best-of-N or MCTS-based methods).
>
> The most important comparison method is PKPO, so we focused our analysis on this method. PKPO constructs an unbiased gradient estimator for the pass@k metric, so it is a theoretically well-justified method and quite fundamental. Our method has the same unbiasedness property, but it reduces variance compared to PKPO. Therefore, by showing that our method outperforms PKPO, it becomes a preferred method for pass@k optimization (in particular the base model comparisons in the previous response highlight that using the regular PKPO may not actually improve performance, whereas our method does).
>
> >The main motivation of this paper is reducing gradient variance. While gradient variance is successfully reduced, … Could the authors provide further analysis on the relationship between variance reduction and final performance?
>
> Variance reduction is typically well understood to speed up optimization (e.g., see [1]). Regarding the variance analysis in the paper, we show variance analyses on toy tasks in the main paper in figures 1 and 3, showing variance scaling with the number of actions $|\mathcal{A}|$ as well as the number of retries $K$. Both the empirical learning performance in terms of reward and the gradient variance are improved (rewards by more then 2x in figure 3). Regarding LLM tasks, it is more difficult to obtain variance estimates due to computational difficulties, so we opted for estimating the variance from the momentum variables stored in Adam, and the LLM gradient variance is reduced as shown in figure 6. Additionally, figures 7, 8 and 9 in the appendix show more results on toy tasks with different settings. We have worked hard on including as much variance analysis as we could, but are open to considering extra analyses if you could specify what type you had in mind.
>
> >Under what conditions does the proposed L2O baseline provide the largest benefits. … (in terms of group size B, objective size K, reward sparsity, or task difficulty, etc.).
>
> Please see figure 1 in the main paper and 7, 8, 9 in the appendix. The largest improvement from L2O comes when the retry parameter K is small; increasing the action space size $|\mathcal{A}|$ also increases the variance reduction; while the group size $B$ does not have much of an effect. Reward sparsity and task difficulty are interesting, but may be difficult to quantify.
>
> >Additional comparison with other baselines should be provided. Alternatively, approaches such as Best-of-N optimization or search-based methods may be complementary rather than strictly competing. Could the authors discuss whether L2O can be combined with such methods or how it would behave in those settings?
>
> It is a good point that many other approaches are complementary rather than competing. Our method works via a reward transform “new_reward_batch = ei_l2o(old_reward_batch)” that can be implemented via a one line change in the code (as long as the ei_l2o method is implemented). The new_reward_batch can be sensibly used in any other REINFORCE gradient estimator based algorithm to make the algorithm target the pass@k objective. In the current work, we focused on comparing the standard variant, and benchmarking against PKPO, as this removes any bias effects and keeps the method clean. However, it can also be plugged into other algorithms, such as the Best-of-N method [2] that essentially adds off-policy sampling to PKPO. We will add more of such discussions.
>
> [1] Mohamed, Shakir, et al. "Monte carlo gradient estimation in machine learning." Journal of Machine Learning Research 21.132 (2020)
>
> [2] Bagirov, Farid, et al. "The Best of N Worlds: Aligning Reinforcement Learning with Best-of-N Sampling via max@ k Optimisation." arXiv preprint arXiv:2510.23393 (2025).

---

> > ### Author Rebuttal · Reviewer_sKfN · 2026-04-08
> >
> > Thank you for your response.
> > I will maintain my score, as I remain inclined toward acceptance.

---

### Decision · Program_Chairs · 2026-04-30

**Decision:**

Reject

**Comment:**

The reviewers agree that the paper is clearly written and that the core theoretical idea is principled. The toy experiments cleanly validate the theory. While the reviews are generally positive on the theory, I do not think the formal novelty and technical positioning/contextualization in the literature were fully stress-tested in the discussion.

At the same time, I share several concerns raised across the reviews, and I do not think the rebuttal fully resolved them.

A first issue is the narrowness of the comparative evaluation. The LLM experiments compare against GRPO, Entropy-Adv, and PKPO. While PKPO is the most natural direct baseline, the paper cites but does not compare against other pass@K algorithms in the literature, for example Chen et al. (2025b), which even proposes a dedicated algorithm for balancing pass@1 and pass@K in the same binary-reward setting used here. Given that the pass@1 degradation is acknowledged as a limitation of pass@K methods, the absence of this comparison is a notable gap. The authors argue in rebuttal that outperforming PKPO is sufficient since it is the closest unbiased estimator, but this sidesteps the question of how L2O compares to methods that address the same practical tradeoffs from different angles. Other concurrent pass@K methods are also not cited or discussed in depth. There is a slight nuance that most of these are derived for binary rewards, but one could in principle apply them more generally, and in any case the paper's LLM experiments are in the binary setting.

If the intended distinction is instead that the proposed method targets the more general continuous-reward max@K setting, then the paper should make this application scope much clearer and provide more direct discussion, analysis, or experiments clarifying the benefit of continuous rewards relative to binary rewards (and the existing works on it). As written, the paper positions itself broadly as a pass@K method, but does not sufficiently compare to, analytically situate itself relative to, or even discuss several recent binary pass@K approaches. I recommend including such references and discussions, if not direct comparisons.

A second issue is the empirical scope. a concern raised by multiple reviewers including the most supportive one is that improvements are consistent but modest in absolute terms. The evaluation covers only verifier-based math reasoning at two model scales (3B and 7B), with a rebuttal addition of Qwen3-4B.  The paper would benefit from clearer characterization of the practical regime where L2O provides the most value. I found the response to reviewer sKfN's question about when the method provides the largest gains too terse and not sufficiently informative.

Finally, the paper lacks a limitations section. While Section 5.3 briefly observes that increasing K lowers pass@1, a candid discussion of this dynamic as a limitation of the method is largely absent from the paper and only directly addressed in the rebuttal. The relationship to methods that specifically address this tradeoff (e.g., Chen et al., 2025b) is not examined. More broadly, related to the first point above, I believe the paper does not sufficiently clarify when practitioners should prefer L2O-based optimization over alternatives with different tradeoffs.

I see merit in the contribution and believe the L2O baseline is a clean addition to the pass@K literature. However, the empirical case is narrower than the paper's positioning suggests, both in baselines and settings. Recognizing this is borderline, I recommend weak reject for the reasons discussed above. I encourage the authors to broaden the experimental comparison and related work as well as add a more candid treatment of scope and limitations in a revision.